# Bidirectional communication between nucleotide and substrate binding sites in a type IV multidrug ABC transporter

Victor Hugo Pérez Carrillo[1,8], Margot Di Cesare [2,8], Dania Rose-Sperling [1], Waqas Javed [2], Hannes Neuweiler [3], Julien Marcoux [4,5], Cédric Orelle [2,9] ✉, Jean-Michel Jault [2,9] ✉ & Ute A. Hellmich [1,6,7,9] ✉

ATP-binding cassette (ABC) transporters use ATP to transport substrates across membranes. In type IV ABC transporters, which include many multidrug resistance (MDR) pumps, communication between nucleotide-binding domains (NBDs) and transmembrane domains (TMDs) is mediated via large intracellular domains containing 'coupling helices'. However, how ATP hydrolysis and substrate transport are functionally coordinated remains unclear. In the bacterial type IV MDR transporter BmrA, we identify a conserved residue cluster at the NBD/TMD interface centered on W413. Mutation of this tryptophan uncouples ATP hydrolysis from transport activity. Mutagenesis, functional assays, nuclear magnetic resonance spectroscopy, hydrogen-deuterium exchange mass spectrometry, and photo-induced electron-transfer fluorescence correlation spectroscopy show that the cluster forms a bidirectional communication hinge that relays signals between the NBD and TMD via coupling helix 2. Hinge mutations affect both local and global dynamics thereby influencing transporter activity. These findings uncover an allosteric pathway critical for functional coupling in multidomain ABC transporters.

ABC transporters are one of the largest protein superfamilies present in all kingdoms of life. They transport chemically diverse substrates including lipids, ions, peptides and vitamins across cellular membranes at the expense of ATP hydrolysis[1,2]. All ABC transporters share a common architecture composed of two transmembrane domains (TMDs) responsible for substrate binding and translocation as well as two cytoplasmic nucleotide-binding domains (NBDs)[2,3]. The four domains can be fused on a single polypeptide ('full transporter') or

assembled e.g. by two halves each comprising a TMD and an NBD ('half transporter') that then dimerize to form a functional unit. ABC transporter NBDs are the most conserved regions regarding both sequence and structure and contain the motifs responsible for ATP binding and hydrolysis, such as the Walker A and Walker B motifs[4–6]. In contrast, the TMDs are structurally and sequentially diverse across the ABC superfamily, resulting in the recent classification into seven subfamilies[3].

[1]Faculty of Chemistry and Earth Sciences, Institute of Organic Chemistry and Macromolecular Chemistry, Friedrich Schiller University Jena, Humboldtstraße 10, 07743 Jena, Germany. [2]Molecular Microbiology and Structural Biochemistry (MMSB), UMR 5086 CNRS/University of Lyon, Lyon, France. [3]Department of Biotechnology & Biophysics, Julius-Maximilians-University Würzburg, Am Hubland, 97074 Würzburg, Germany. [4]Institut de Pharmacologie et de Biologie Structurale (IPBS), Université de Toulouse, CNRS, Université de Toulouse, Toulouse 31077, France. [5]Infrastructure Nationale de Protéomique, ProFI, UAR, 2048 Toulouse, France. [6]Cluster of Excellence "Balance of the Microverse", Friedrich Schiller University Jena, 07743 Jena, Germany. [7]Centre for Biomolecular Magnetic Resonance (BMRZ), Goethe University, Max von Laue Str. 9, 60438 Frankfurt, Germany. [8]These authors contributed equally: Victor Hugo Pérez Carrillo, Margot Di Cesare. [9]These authors jointly supervised this work: Cédric Orelle, Jean-Michel Jault, Ute A. Hellmich. ✉e-mail: cedric.orelle@cnrs.fr; jean-michel.jault@cnrs.fr; ute.hellmich@uni-jena.de

Following this classification, many multidrug resistance (MDR) pumps such as BmrA from *B. subtilis*[7] or mammalian P-glycoprotein[8] (P-gp, ABCB1) belong to the type IV subfamily of ABC transporters. During their catalytic cycle, these transporters alternate between inward-facing (IF) and outward-facing (OF) conformations to mediate drug export from the cell (Supplementary Fig. 1A). In the absence of nucleotides (apo state) the IF is more populated, whereas ATP binding and subsequent NBD dimerization shift the equilibrium towards the OF state, triggering conformational changes in the TMD that facilitate drug release. The OF can also be stabilized by vanadate which, following ATP hydrolysis, traps ADP-Mg$^{2+}$ in the nucleotide-binding sites (ADP*Vi state)[9]. In type IV transporters, interdomain crosstalk between NBD and TMD is mainly attributed to the close spatial arrangement of the conserved Q–[10,11] and X–loop motifs[12–14] in the NBDs with two so-called coupling helices[15–18] in the TMDs. In half transporters like BmrA, which contains six transmembrane helices per protomer, these coupling helices are part of two large intracellular domains 1 (ICD1) and 2 (ICD2) linking transmembrane helices 2-3 and 4-5, respectively[19] (Supplementary Fig. 1). Upon dimerization into the functional transporter, ICD1 contacts the NBD within the same protomer (in *cis*), while ICD2 reaches over to the NBD of the opposing subunit (in *trans*), inserting into a groove on the NBD surface between the RecA and the α-helical subdomains[13,20]. This 'swapped topology' is the hallmark of the type IV ABC subfamily[3,13] (Fig. 1A, Supplementary Fig. 1A).

Since ABC transporters efficiently couple ATP hydrolysis with substrate translocation, it is crucial to integrate the existing architectural framework of type IV ABC transporters with a long-range dynamic coupling network that facilitates effective crosstalk between the NBD and TMD. A vast body of work highlighted the importance of the interdomain contacts mediated by the ICDs for the ABC transporter catalytic cycle, including mutagenesis and crosslinking experiments, biophysical and structural as well as computational methods, e.g. refs. 13,16–18,21–25. However, a precise mechanistic understanding of interdomain crosstalk in ABC transporters remains currently amiss, specifically regarding the dynamics of the molecular dialogue between nucleotide and substrate interaction sites that are typically >40 Å apart.

Conveniently, the isolated NBDs of bacterial type IV ABC transporters remain monomeric in solution, while preserving their native fold and the ability to interact with nucleotides. This makes them well-suited as model systems for high-resolution studies using solution nuclear magnetic resonance (NMR) spectroscopy[26–28]. We previously observed that tryptophan W421 in the isolated NBD of the *L. lactis* MDR type IV ABC transporter LmrA, ~22 Å away from the nucleotide binding site and upstream of the Q-loop, senses nucleotide binding in NMR chemical shift perturbation experiments[26]. Furthermore, Trp fluorescence was shown to be a sensitive reporter of drug binding to the TMD in the homologous *B. subtilis* transporter BmrA[7] and we hypothesized that this was due to the corresponding tryptophan in the NBD (W413). Together this suggested that a conserved tryptophan in the NBD of type IV MDR transporters might be part of a previously unrecognized long-range sensing network. Importantly, this Trp residue is located close to the tip of ICD2 (Fig. 1A), pointing to a role as an allosteric sensor in the NBD/TMD interface capable of integrating nucleotide and substrate signals.

Here, combining mutagenesis, functional assays, solution $^1$H$^{15}$N and $^{19}$F NMR spectroscopy, photo-induced electron transfer fluorescence correlation spectroscopy (PET-FCS) and hydrogen deuterium exchange coupled to mass spectrometry (HDX-MS) on the bona fide MDR ABC transporter BmrA[7], a prototypical type IV ABC member[29,30], we identified a previously uncharacterized communication hinge in the nucleotide-binding domain comprising three conserved residues, with W413 at its center. In addition to maintaining structural stability, this hinge forms a bidirectional dynamic relay to transmit information on nucleotide and drug interaction between NBD and TMD via ICD2.

Our findings reveal a pathway for interdomain crosstalk in a type IV ABC transporter and provide insights into the intricate mechanisms of allosteric interdomain coupling in this transporter subfamily.

## Results

### A conserved triad of residues in Type IV ABC transporters NBDs senses remote nucleotide binding

ABC transporters use ATP to power substrate translocation. The nucleotide interacts with conserved motifs in the NBD, including a lysine residue (K380 in BmrA) from the Walker A motif that assists hydrolysis[31] (Fig. 1A). Nucleotide binding is thought to be relayed to the NBD/TMD interface through another conserved motif, the Q-loop, although the precise molecular pathway remains unknown[10,11]. A sequence alignment of NBDs from type IV ABC transporters showed that the site we previously found to be sensitive to nucleotide or drug binding in LmrA[26] and BmrA[7] typically harbors a bulky hydrophobic side chain such as tryptophan (e.g. BmrA W413, LmrA W421) or leucine (e.g. MsbA L415) (Fig. 1B). In bacterial half transporters closely related to BmrA (i.e. UniProtKB sub-database, 226 sequences extracted from the UniRef50_O06967 database), the tryptophan residue is ubiquitously present.

To elucidate the functional role of this residue and its potential interaction with other sites, we compared the fingerprint $^1$H, $^{15}$N-HSQC solution NMR spectra of the ~29 kDa $^{15}$N-labeled NBDs of three homologous bacterial type IV transporters: the multidrug transporters *B. subtilis* BmrA and *L. lactis* LmrA, as well as the *E. coli* lipid A transporter MsbA in the apo and in the presence of ATP and ADP (Fig. 1C, Supplementary Fig. 2). To this end we took advantage of our previously published and de novo determined NMR backbone assignments for the three proteins (BMRB entries 17660[27], 51156[28], and 52626). In all cases, more than 91% of all non–proline residues were successfully assigned (Supplementary Fig. 2D).

In all three NBDs, the chemical shift of the abovementioned hydrophobic residue (BmrA W413, LmrA W421, MsbA L415) responded to the addition of ATP or ADP (Fig. 1C). In the case of the tryptophan residues in BmrA and LmrA, a shift in both the backbone and sidechain amide resonances was observed. To reflect the chemical nature of the side chain, we denote this residue as φ. It resides in a short α-helix within the RecA subdomain and is consistently followed by a strictly conserved arginine (BmrA R414, LmrA R422, MsbA R416), which points towards the coupling helix from ICD2 of the *trans* protomer in these half transporters (Fig. 1A, zoom). Facing residue φ, at the C-terminal end of the helix that directly follows the Walker A motif, lies another highly conserved arginine residue (BmrA R389, LmrA R397, MsbA R391). Both arginine residues will be referred to hereafter as R$^{ICD2}$ and R$^{WA}$ to mark their respective location. Residue R$^{WA}$ showed a marked chemical shift change upon addition of ATP or ADP, while the effects were very subtle for R$^{ICD2}$ (Fig. 1C).

Together, R$^{WA}$, φ and R$^{ICD2}$ appear to be ideally positioned to serve as a 'communication hinge' that allosterically links the nucleotide binding site to the NBD/TMD interface. This connection likely extends from the Walker A motif and the following helix through R$^{WA}$, to φ, and into R$^{ICD2}$ reaching the TMD (Fig. 1A).

### The communication hinge is important for protein stability and transporter function

The communication hinge plays a role in both nucleotide sensing and structural integrity. Circular Dichroism (CD) spectroscopy and analytical size exclusion chromatography (SEC) revealed that point mutations in the corresponding residues in BmrA, LmrA and MsbA destabilize the isolated NBDs, with the most pronounced effects for mutations of residue R$^{WA}$ (Supplementary Fig. 4). Notably, substitutions of BmrA W413 with leucine and alanine could not be purified in the context of the isolated NBD. Furthermore, the melting temperatures ($T_m$) of the hinge mutant NBDs that could be obtained were

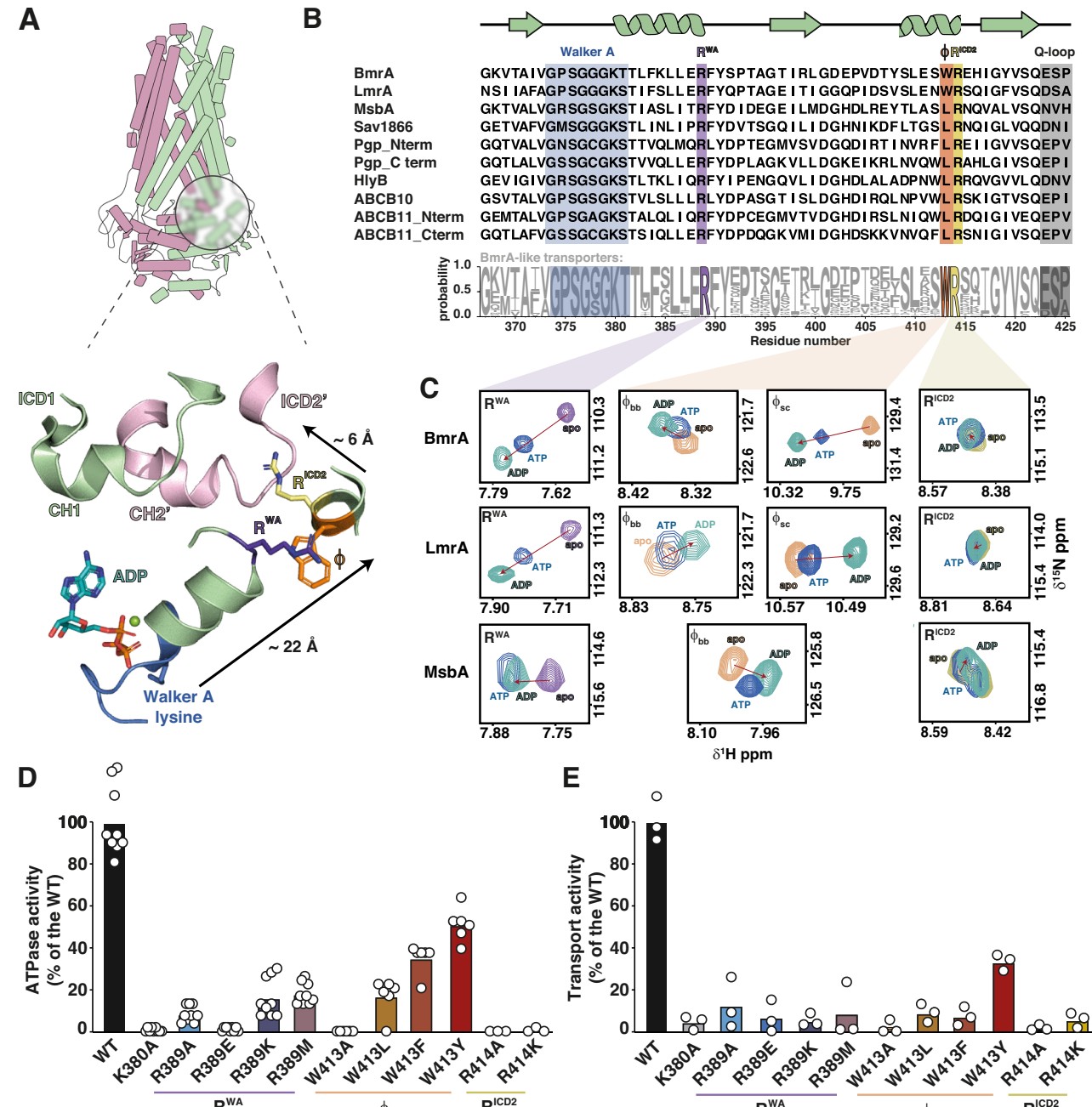

**Fig. 1 | The communication hinge senses nucleotide binding and is crucial for activity. A** Structure of *B. subtilis* BmrA (PDB ID: 6R81[29]), an archetypical type IV ABC half transporter (protomers shown in pink and green). The zoom shows the Walker A motif (blue) and the Walker A helix leading into the 'hinge' consisting of a conserved arginine residue in the Walker A helix (R$^{WA}$, purple), a conserved bulky hydrophobic residue (φ, orange) and a conserved arginine pointing towards intracellular domain 2 (R$^{ICD2}$, yellow) (Structural overlay of this region in other ABC transporters shown in Supplementary Fig. 2A). **B** High degree of conservation within the Walker A and hinge residues (purple, orange and yellow) in both human and bacterial type IV ABC transporters (top). Residue numbers are based on the sequence of *B. subtilis* BmrA. Note that in bacterial transporters with at least 50% identity with BmrA, residue φ is a strictly conserved Trp residue (based on 226 type IV NBD sequences, bottom). Sequence logo was created using WebLogo3[64]. **C** Nucleotides are sensed by the communication hinge residues. Comparison of NMR spectra of ¹⁵N-labeled isolated NBDs of BmrA, LmrA and MsbA in the apo state (purple, orange or yellow resonances, respectively) and in the presence of 10 mM

ATP (blue) or ADP (teal) reveals chemical shift perturbations of the backbone (and in the case of tryptophan also side chain, φ$_{sc}$) amide resonances of the three hinge residues. Shown is a zoom into the respective ¹H, ¹⁵N-HSQC NMR spectra (Supplementary Fig. 2C) to highlight the three hinge residues. **D, E** The hinge residues are crucial for transporter function. ATPase activity of full-length BmrA variants reconstituted in MSP1E3D1 nanodiscs prepared with *E. coli* total lipid extract (**D**) and fluorescence-based transport assay with doxorubicin in inside-out vesicles prepared from *E. coli* cells overexpressing BmrA variants (**E**). In both cases, values were normalized to the WT set as 100%. For ATPase activity of WT, K380A and R389X constructs, results shown are the mean of three biological triplicates with three technical replicates each. For W413X constructs, results shown are the mean of two biological replicates with three technical replicates each. For R414X constructs, results shown are the mean of one biological replicate with three technical replicates. For transport activity of all constructs, results shown are the mean of three technical replicates. Protein expression and folding were not affected by the mutations (Supplementary Fig. 3).

consistently lower than the respective wildtype (WT) proteins (Supplementary Table 1), although nucleotide dissociation constants were only slightly altered for most cases (e.g. $K_{D, ADP} = 214 \pm 43$, $221 \pm 105$, $221 \pm 54$ and $266 \pm 97$ µM for BmrA NBD WT, R389K, W413F and R414K, respectively). In contrast, more drastic substitutions such as R389M and R414A reduced the affinity of nucleotides two- and ten-fold, respectively (Supplementary Table 2).

To explore the role of the communication hinge for transporter function, point mutations were introduced into full-length *B. subtilis* BmrA to yield the respective BmrA $R^{WA}$ (R389A/K/E/M), φ (W413 A/F/Y/L) and $R^{ICD2}$ (R414A/K) variants. As a negative control for functional assays, the Walker A lysine mutant BmrA K380A[32] was also prepared. The successful expression and purification of all hinge variants in detergent micelles suggests that the TMD can partially mitigate folding defects that were pathological for the isolated NBD (Supplementary Fig. 3A-C). However, like in the isolated NBDs, a consistent decrease in the $T_m$ for the hinge variants was observed in the full-length transporter (Supplementary Table 3). Notably, while the WT and most of the communication hinge mutants were stabilized by nucleotides and could be trapped in the OF conformation by ADP*V$_i$, both the R414A and R414K substitutions as well as the K380A mutant showed no increase in $T_m$ in the presence of nucleotides. This suggested that they cannot reach the outward facing (OF) state[33] (Supplementary Table 3).

The ATPase activity of purified BmrA full-length WT, K380A and hinge mutant constructs was determined in detergent micelles, lipid nanodiscs and liposomes (Fig. 1D, Supplementary Fig. 3D, E). Transport activity was measured in inside-out vesicles prepared from *E. coli* cells overexpressing BmrA variants using the fluorescent drug doxorubicin (Fig. 1E; Supplementary Fig. 3F, G). Mutations in the hinge severely impaired both ATPase activity and transport, even for conservative substitutions. Mutation of the $R^{ICD2}$ residue to alanine or lysine completely abolished function. $R^{WA}$ and φ residue mutants showed a more graded response: The R389A/E and W413A mutants showed very little to no activity, while the R389K/M and W413L mutants displayed some and the W413F/Y mutants retained substantial residual ATPase activity. Intriguingly, the W413F and W413Y mutants appeared functionally uncoupled, with ATPase activity much less affected than doxorubicin transport. Consistent with our findings, W413 mutations in BmrA were also recently reported to differentially impair Hoechst 33342 transport (W413A > W413F > W413Y), although the ATPase activities of the mutants were not measured in that study[34].

Our results show that mutations in the communication hinge, even conservative ones, severely impair both ATPase and transport activity. Notably, mutations of W413 can also uncouple these two functions, suggesting that this residue acts as a linchpin, integrating the signals that coordinate ATP hydrolysis with substrate transport.

## Nucleotide binding site and communication hinge are bidirectionally coupled

Given that residues $R^{WA}$, φ and $R^{ICD2}$ can remotely sense nucleotide binding and regulate BmrA ATPase activity, we hypothesized that the three communication hinge residues may be structurally coupled to each other and allosterically linked to the nucleotide binding site (Fig. 2A). To test this, we used isolated NBD constructs bearing hinge residue mutations that were sufficiently thermostable for NMR spectroscopy (Supplementary Table 1). One-dimensional $^{19}$F-NMR and 2D $^1$H, $^{15}$N-correlation spectra showed that single point mutations at any of the hinge positions induced chemical shift perturbations and, in some cases, line broadening at the other two sites (Fig. 2B–D, Supplementary Fig. 5). For example, substituting $R^{WA}$ with either methionine or lysine led to the complete loss of the indole side chain NH signal from W413 in the $^1$H, $^{15}$N HSQC spectra of the apo state (Supplementary Fig. 5C). These observations suggest both structural and dynamic local coupling within the hinge.

To nonetheless obtain information about the W413 sidechain, we used $^{19}$F NMR (Fig. 2B). Fluorine labeling of tryptophan residues can be easily achieved by feeding the labeled amino acid to the expression host[35–37]. Conveniently, W413 is the only native tryptophan residue in the BmrA NBD, yielding a single resonance in the 1D $^{19}$F NMR spectrum (Fig. 2B, black). Of note, $^{19}$F NMR measurements could only be carried out for the $^{19}$F-Trp labeled WT NBD and the $R^{WA}$ point mutants R389K and R389M (Fig. 2B, blue, mauve), since $R^{ICD2}$ mutants became unstable with the fluorine label and could not be purified. Mutation of $R^{WA}$ led to line broadening and a strong chemical shift of the fluorinated W413 indole ring, suggesting altered tryptophan sidechain dynamics. Importantly, the fluorinated protein retained its sensitivity to nucleotides as seen by the changes in $^{19}$F chemical shift upon ADP addition (Fig. 2B top versus bottom spectra). Moreover, the introduction of the Walker A lysine mutant K380A mutation, more than 15 Å away from W413 (see Fig. 1A), also induced a chemical shift change in the W413 $^{19}$F resonance (Fig. 2B, bottom). This strongly supports the idea of direct crosstalk between the communication hinge and the nucleotide binding site.

Interestingly, in the $^1$H, $^{15}$N-HSQC spectra of the apo-state WT NBD, most backbone amide signals belonging to the Walker A motif, including K380, are severely broadened[28] (Supplementary Fig. 2B). Upon nucleotide binding, these signals become resolved, suggesting that the Walker A motif and adjoining helix are stabilized. Taking advantage of the ADP-bound state, which allows clear observation of the Walker A backbone resonances, we next examined whether mutations in the communication hinge are sensed by the Walker A lysine K380 (Fig. 2C). In fact, mutation of any of the three hinge residues led to chemical shift changes or severe line broadening for the K380 backbone amide resonance, shown in Fig. 2C both in the 2D spectrum and in the 1D projection. The most pronounced effects on K380 resulted from mutation of R389 ($R^{WA}$), followed by substitutions in W413 (φ) and R414 ($R^{ICD2}$) (see also Supplementary Fig. 6). Reciprocally, mutation of the Walker A residue to alanine also resulted in chemical shift perturbations of the backbone (and in the case of W413 also the sidechain) NH chemical shifts and line widths of all three hinge residues (Fig. 2D, Supplementary Fig. 7). Despite the reduced affinity of the K380A mutant for ADP (Supplementary Table 2), this effect was even more pronounced in the presence of 10 mM nucleotide compared to the apo state. Overall, this supports a crosstalk from hinge to Walker A motif, and vice versa.

## Nucleotides and mutation of the communication hinge modulate NBD subdomain dynamics

ABC transporter NBDs are composed of an α-helical and a RecA-like subdomain, whose dynamics are modulated by nucleotide binding[38]. Coupling helix 2, which works together with coupling helix 1 to mediate NBD/TMD communication, nestles into a groove formed by the two NBD subdomains, directly on top of the communication hinge (Fig. 3A). We hypothesized that nucleotide-induced changes in the communication hinge might alter the relative dynamics between the RecA- and α-helical subdomains, potentially serving as a mechanism to transmit the signal of ATP binding to the TMD. To test this, we used photoinduced electron transfer fluorescence correlation spectroscopy (PET-FCS). PET-FCS detects and directly measures the reconfiguration time constants of specific protein structural elements, i.e. structural fluctuations of biomacromolecules on the msec to µsec timescale, by monitoring quenching of a fluorophore by a nearby tryptophan sidechain[39,40]. Unlike FRET, PET-FCS therefore requires only a single fluorophore, which avoids complications associated with dual labelling.

To label the BmrA NBD for PET-FCS, we first generated a construct lacking the one native cysteine residue, BmrA NBD C436S. This mutation does not affect BmrA function[30]. A newly introduced cysteine residue, S516C in the RecA subdomain, was then covalently labeled

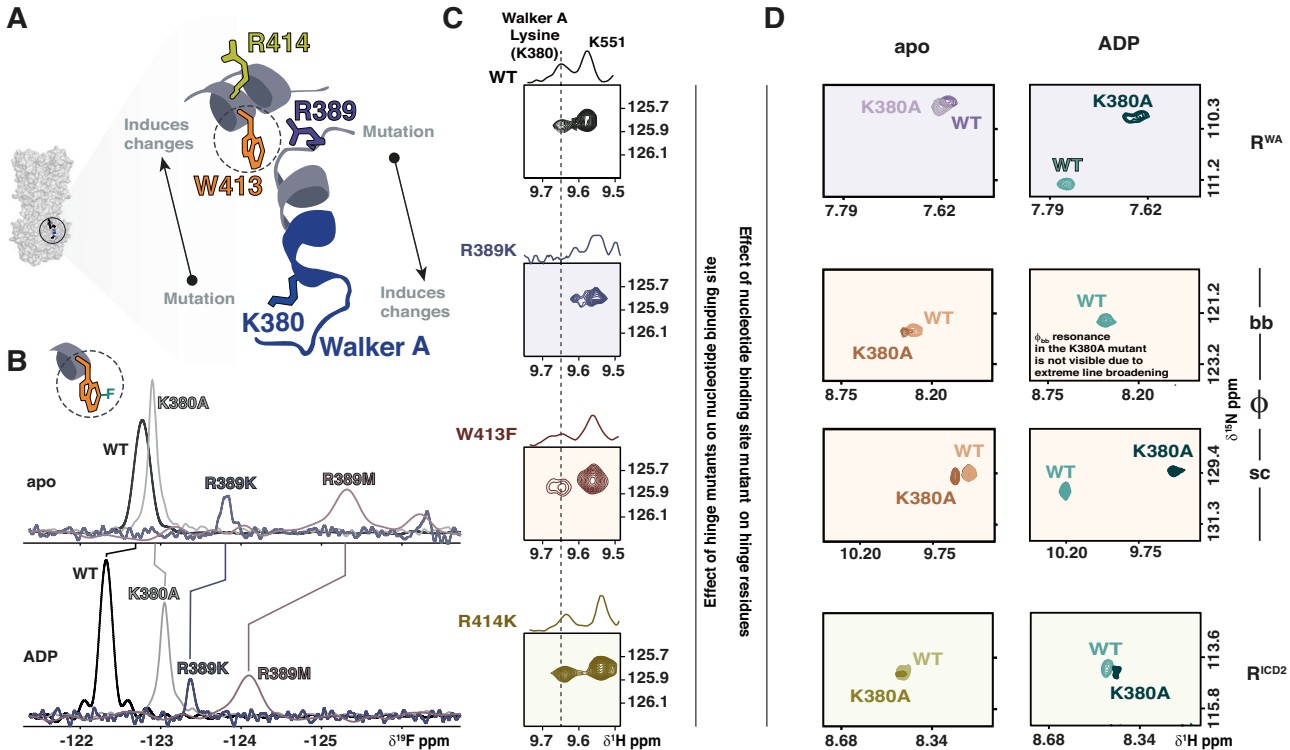

**Fig. 2 | Bidirectional coupling between communication hinge and nucleotide binding site in the BmrA NBD. A** Zoom in into the NBD of BmrA (PDB ID: 6R81[29]) showing the bidirectional crosstalk between the nucleotide binding site and hinge was investigated by introducing mutations at either site and monitoring the effects by $^1H$, $^{15}N$ 2D NMR and $^{19}F$ NMR. **B** 1D $^{19}F$ NMR spectra of $^{19}F$-5Trp-labeled BmrA NBD variants in the absence (top) or presence (bottom) of ADP. Residue φ, W413, is the only native tryptophan in the BmrA NBD, thus giving rise to a single fluorine resonance in the $^{19}F$ NMR spectrum of WT BmrA NBD (black). Mutation of the $R^{WA}$ residue into lysine (R389K, blue) or methionine (R389M, mauve) or the Walker A lysine K380 to alanine (grey) led to chemical shift changes and line broadening of W413 (φ), suggesting an influence on the φ sidechain dynamics. Mutations of residue $R^{ICD2}$ (R414A, R414K) led to fully aggregated protein and were thus not included. Spectra of BmrA NBD WT (250 μM), K380A (300 μM), R389K (100 μM) and R389M (250 μM) constructs were recorded at 298 K at 600 MHz with 512, 256, 4096 and 10240 scans, respectively. **C** Hinge mutations are sensed by the conserved Walker A lysine, K380. Shown is a zoom into the $^1H$, $^{15}N$-HSQC and the corresponding 1D projection for the K380 resonance in the WT, R389K, W413F and R414K constructs. In the apo state, the resonance of K380 shows severe line broadening, thus experiments were carried out with 10 mM ADP. **D** Mutation of the Walker A lysine is sensed by the hinge residues. Depicted are zooms into the $^1H$, $^{15}N$-HSQC spectra of the WT and K380A BmrA NBD in the absence (left) and presence of 10 mM ADP (right) monitoring the backbone NH resonances of residues $R^{WA}$ (top) and $R^{ICD2}$ (bottom). For residue φ, both the backbone (bb) and sidechain (sc) NH resonances are shown (middle).

with an ATTO Oxa11 maleimide fluorophore. Combined with an engineered tryptophan residue in the α-helical domain, N459W, this allowed to monitor conformational fluctuations between the two subdomains (Fig. 3B, Table 1). As a control, we also measured the fluorescently labeled construct lacking the tryptophan quencher (Fig. 3B, gray trace). Here, the corresponding autocorrelation function (ACF) was well described by diffusion of a globular protein through the confocal detection focus that was void of fluorescence fluctuations in the sub-millisecond time regime. The observed, additional 8 μs relaxation had a negligible amplitude of only 5%. This confirmed that all detected changes by FCS in the construct containing the fluorophore/tryptophan reporter pair originate from quenching interactions between the engineered Trp at position N459W and the Atto Oxa11 label. These fluctuations directly reflect relative motions between RecA and α-helical subdomains. The resulting ACF exhibited three single-exponential relaxations in the sub-millisecond range, each with substantial amplitude (Fig. 3B, black trace). While it is difficult to assign specific relaxations to specific motions, the data indicates three modes of motions of the two NBD subdomains relative to each other on the μs-ns time scale, which may include shearing, bending or breathing.

Both nucleotide binding and mutation of the hinge residue φ (W413F) modulated these subdomain dynamics (Fig. 3C). Addition of ADP accelerated the 80 μs mode of motion to 24 μs, while binding of

ATP slowed the motion to 253 μs (Table 1). In the W413F mutant, intradomain NBD dynamics were altered substantially (Fig. 3B, C, brown traces). One mode of motion, i.e., the 2.5 μs relaxation, was abolished and the amplitude of the ns relaxation was considerably reduced, while the 80 μs relaxation was accelerated. In the mutant, nucleotide binding slowed the 39 μs relaxation to time constants >100 μs and the 0.57 μs motion to 2.7 and 5.7 μs with ATP and ADP, respectively (Fig. 3C, Table 1). Together, the changes observed in the PET-FCS measurements between the apo and nucleotide-bound states, as well as between WT and mutant NBDs, clearly demonstrate that both nucleotide binding and mutation of the communication hinge alter the dynamic coupling between the RecA and α-helical subdomains of the NBD.

### The communication hinge impacts global transporter dynamics through ICD2

Our PET-FCS results showed that the communication hinge plays a role in NBD dynamics between RecA and α-helical subdomain. Since coupling helix 2 sits in the groove between these two subdomains, we hypothesized that the communication hinge could also influence TMD dynamics via ICD2. Hydrogen–deuterium exchange coupled to mass spectrometry (HDX–MS) and $^{19}F$ NMR spectroscopy are powerful tools for studying conformational dynamics of multidomain membrane proteins in lipid-like environments[25,35,41,42]. Here, we

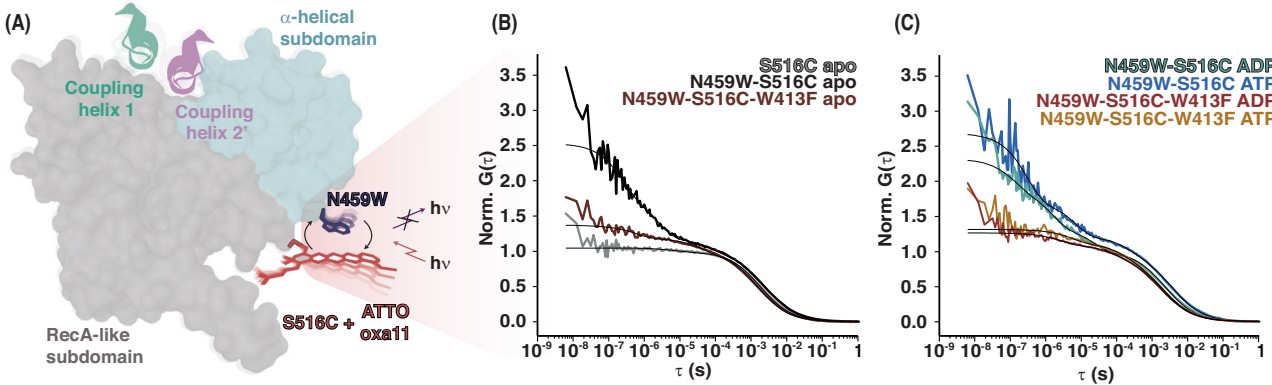

**Fig. 3 | Nucleotide binding and hinge mutations modulate the dynamics between the RecA-like and α-helical subdomains of the NBD. A** Photo-induced electron transfer fluorescence correlation spectroscopy (PET-FCS) was used to detect intradomain structural rearrangements between RecA and α-helical subdomains in the NBD. An ATTO Oxa11 maleimide fluorophore was attached to an engineered cysteine in the RecA-like subdomain (S516C), where the native cysteine was removed (BmrA NBD C436S), in vicinity to a newly introduced tryptophan in the α-helical domain (N459W) serving as the quencher for the fluorophore.

**B, C** PET-FCS autocorrelation functions of fluorescently labeled BmrA NBD constructs were recorded in the absence (**A**) or presence (**B**) of nucleotides. PET-FCS reporters were designed to monitor fluctuations between α-helical and RecA subdomains. In the cysteine-less background, the fluorescently modified S516C, S516C-N459W and S516C-N459W-W413F served as a control, a reporter for hinge motions, and a probe for the role of conserved W413, respectively. Black lines are fits to the data using a model for a single diffusion containing one, two or three single-exponential relaxations.

combined these methods on the full-length BmrA WT transporter and the W413F mutant to investigate interdomain communication between the NBD and TMD in the apo and ADP*$V_i$ trapped states, mainly populating the inward facing (IF) and outward facing (OF) conformations, respectively (Fig. 4, Supplementary Table 4, Supplementary Fig. 8).

Overall, >90% peptide sequence coverage of both WT and W413F mutant reconstituted in nanodiscs could be achieved in the HDX experiments (Supplementary Fig. 8). In line with previous studies[33,43], the NBDs had better peptide coverage than the TMDs, which are partially shielded from proteolysis by the membrane scaffold protein (MSP)-lipid belt. The deuterium uptake plots for all the peptides of the WT and the W413F mutant (Supplementary Figs. 9, 10, 11) display mainly classic EX2 kinetics. However, some notable exceptions exhibit EX1 behavior, as apparent from the characteristic 'bi-modal' distribution, in ICD1 (between residues 108 to 132), ICD2 (between residues 189 to 255), the linker region between the TMD and the NBD (between residues 304-343) or the α-helical subdomain of the NBD (between residues 480-500) (Supplementary Fig. 12).

In the apo state, HDX of the WT protein suggested significant flexibility, particularly within the NBDs and ICDs (Supplementary Fig. 8C). Upon ADP*$V_i$ trapping, HDX globally decreased in the WT transporter showing rigidification (Fig. 4), a behavior consistently seen across type IV ABC transporters upon nucleotide binding[12,29,44–47]. Deuterium exchange differences between apo and trapped states were most pronounced for the conserved motifs in the NBD (e.g. Walker A and C-loop) and the intracellular domains 1 and 2 (ICD1, ICD2), particularly the C-terminal end of TMH4, leading into ICD2, and TMH5, which transitions from ICD2 into the TMD (Fig. 4A, red regions, Fig. 4B).

In comparison to the WT, the W413F hinge mutant exhibited subdued exchange differences between apo and MgADP*$V_i$ trapped states in key regions, including the Walker A motif, the hinge and ICD2 and the surrounding helices TMH4 and 5 (Fig. 4A, B, Supplementary Fig. 8). Notably, a major consequence of the W413F mutation was increased HDX in the ADP/Vi trapped state compared to the WT, highlighting that nucleotide-dependent interdomain communication between NBD and TMD is primarily mediated via ICD2, and is significantly disrupted when the hinge is mutated.

Because the W413F hinge mutation caused some destabilization of the protein, as reflected by a reduced melting temperature (Supplementary Table 3), it can broadly affect protein dynamics. This is consistent with an increase in deuteration seen with several peptides in the apo state of the mutant (Supplementary Table 5). Therefore, we chose not to directly compare global HDX rates between WT and W413F but, for a focused comparison, we selected six representative shared peptides from the NBD and ICD with comparable deuteration levels in the apo state (Fig. 4C, Supplementary Table 5). These data reveal reduced protection in the ADP*$V_i$ state for the mutant relative to the WT, supporting our hypothesis that W413 plays a key role in signal transmission across the transporter. Of note, peptides within the drug-binding site in the transmembrane domain were either not covered or showed deuteration levels too low for confident interpretation.

## Lipid environment and substrates influence hinge dynamics

To complement the HDX-MS data, which provides information at the peptide level, we used [19]F NMR spectroscopy in solution with the full-length BmrA transporter in detergents and nanodiscs (Fig. 5). In addition to W413 in the NBD, BmrA contains two other native tryptophan residues in the TMD: W104 in ICD1 and W164 located within the extracellular loop between TMH3 and TMH4 (Fig. 5A). These were individually mutated to facilitate [19]F NMR assignments and the structural and functional integrity of the fluorinated transporter was confirmed by CD spectroscopy, transport and ATPase activity assays (Fig. 5B-D). Together with the isolated [19]F-Trp labeled NBD, we unambiguously assigned the three fluorine resonances of [19]F-5Trp labeled BmrA to the two tryptophan residues in the TMD and to the hinge residue W413 in the NBD (Fig. 5E).

Importantly, the fluorine resonances were readily transferable from detergent-solubilized to nanodisc-reconstituted protein (Fig. 5F). While the W413 resonance became narrower, indicating altered NBD dynamics in the lipid environment as reported for other type IV ABC transporters by complementary methods[48,49], it remained distinct from the TMD tryptophan resonances, suggesting that both the global NBD dynamics, as well as local hinge dynamics are impacted by the transporter environment.

Next, we investigated the effect of nucleotide or substrate binding on W413. As seen in the [19]F NMR spectra of the isolated NBD (Fig. 2B), ATP addition led to chemical shift changes and line narrowing of the

**Table 1 | Fitting parameters for PET-FCS**

| BmrA NBD construct | $a_1$ | $\tau_1$ (µs) | $a_2$ | $\tau_2$ (µs) | $a_3$ | $\tau_3$ (µs) |
|---|---|---|---|---|---|---|
| S516C | 0.05 ± 0.01 | 8 ± 2 | -- | -- | -- | -- |
| S516C-N459W | 0.19 ± 0.03 | 80 ± 30 | 0.62 ± 0.06 | 2.5 ± 0.4 | 0.72 ± 0.06 | 0.24 ± 0.04 |
| S516C-N459W-W413F | 0.19 ± 0.01 | 39 ± 5 | -- | -- | 0.18 ± 0.01 | 0.57 ± 0.01 |
| S516C-N459W + ADP | 0.35 ± 0.03 | 24 ± 4 | 0.56 ± 0.04 | 1.5 ± 0.2 | 0.42 ± 0.04 | 0.13 ± 0.03 |
| S516C-N459W + ATP | 0.22 ± 0.08 | 253 ± 16 | 0.51 ± 0.06 | 7 ± 2 | 0.95 ± 0.06 | 0.30 ± 0.04 |
| S516C-N459W-W413F + ADP | 0.18 ± 0.01 | 104 ± 12 | 0.18 ± 0.01 | 2.7 ± 0.2 | -- | -- |
| S516C-N459W-W413F + ATP | 0.19 ± 0.02 | 177 ± 40 | 0.20 ± 0.01 | 5.7 ± 0.7 | -- | -- |

Kinetic parameters obtained for the BmrA NBD single (S516C), double (N459W-S516C) and the triple (N459W-S516C-W413F) mutants in apo and after incubation with ADP or ATP.

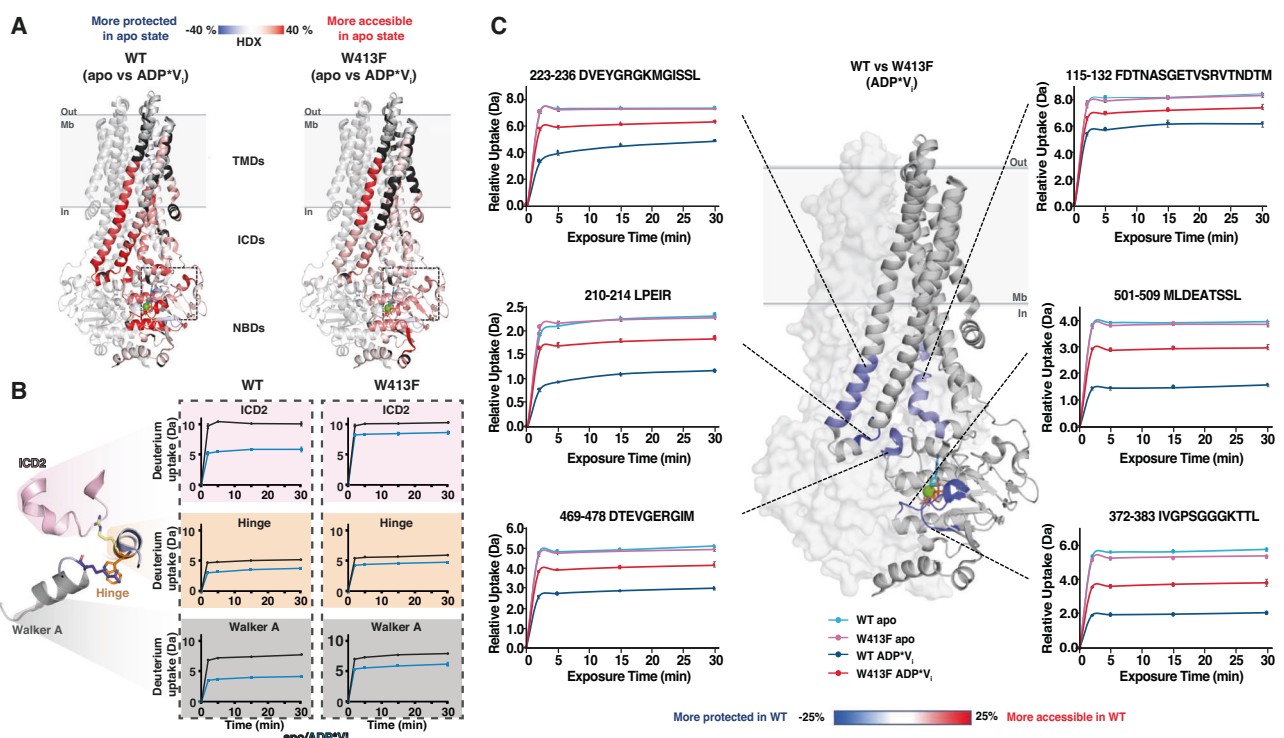

**Fig. 4 | The hinge impacts global transporter dynamics. A** HDX–MS on BmrA WT and BmrA W413F in nanodiscs (Supplementary Tables 4, 5). Differences in HDX after 30 min between apo (IF) and ADP*Vi trapped state (OF, with 10 mM MgATP and 1 mM $V_i$) mapped onto the BmrA cryoEM structure (PDB: 7OW8[29], nucleotide shown as cyan sticks, $Mg^{2+}$ as green sphere). Peptides without significant differences (*p*-value > 0.05) are in light gray and non-covered peptides in black. Regions more accessible in the apo than ADP*Vi state are highlighted in red (sequence coverage and uptake plots in Supplementary Figs. 8,10 and 11). Deuteration uptake presented as mean values of two or three technical replicates ± SEM, for WT or W413F, respectively. Significance was assessed using Peptide-level statistical tests (*p*-value < 0.05). **B** The differential deuterium uptake between the apo and ADP*Vi states is subdued in the vicinity of the hinge upon introduction of the W413F mutation. Zoom of the hinge region highlighting differences in HDX between WT (left) and hinge mutant (right). Selected significant peptides (*p*-value < 0.05) from ICD2 (amino acid residues 216 – 236), the hinge (amino acid residues 413 – 427) and the Walker A helix (amino acid residues 363 – 383) show reduced HDX in the ADP*Vi trapped (blue) compared to the apo state (black). These differences are stronger for the WT than the hinge mutant (see Supplementary Fig. 9). **C** Comparison of deuteration uptake between WT and W413F in nanodiscs for selected peptides with similar deuteration levels in the apo state (Supplementary Table 5). Relative HDX differences after 30 min between the WT and W413F in the MgADP*Vi trapped state (*p*-value < 0.05) are mapped onto the BmrA cryoEM structure in the OF state. Regions more protected in WT than W413F are shown in blue, while other protein regions are gray. One monomer is displayed as a ribbon and the second as a transparent surface.

W413 $^{19}$F NMR signal in the full-length transporter (Fig. 5F). Substrate addition led to severe line broadening, showing that nucleotides and substrates directly affect hinge dynamics in the full-length transporter (Fig. 5F). Of note, due to the changes in pH associated with adding a saturating amount of doxorubicin to the NMR samples, which led to sample precipitation, we instead used reserpine, another BmrA substrate[7,12]. To nonetheless obtain information on the transporter dynamics with doxorubicin, we carried out HDX-MS in the presence of doxorubicin, achieving similar peptide coverage as for the apo and ADP*$V_i$ trapped transporter (Supplementary Fig. 13). Comparison of

BmrA WT in the absence and presence of doxorubicin showed decreased exchange in the TMD, notably TMH4, 5 and ICD2 in the presence of the drug. This shows that substrate binding decreases dynamics in regions that also become rigidified upon nucleotide addition (Fig. 4A). A similar trend was seen for the W413F hinge mutant, although generally substrate-evoked effects were noticeably subdued compared to the WT (Supplementary Fig. 13), suggesting reduced coupling as also apparent from the activity assays (Fig. 1D, E).

In summary, HDX-MS and $^{19}$F NMR demonstrate that both nucleotide binding to the NBD and drug binding to the TMD influence

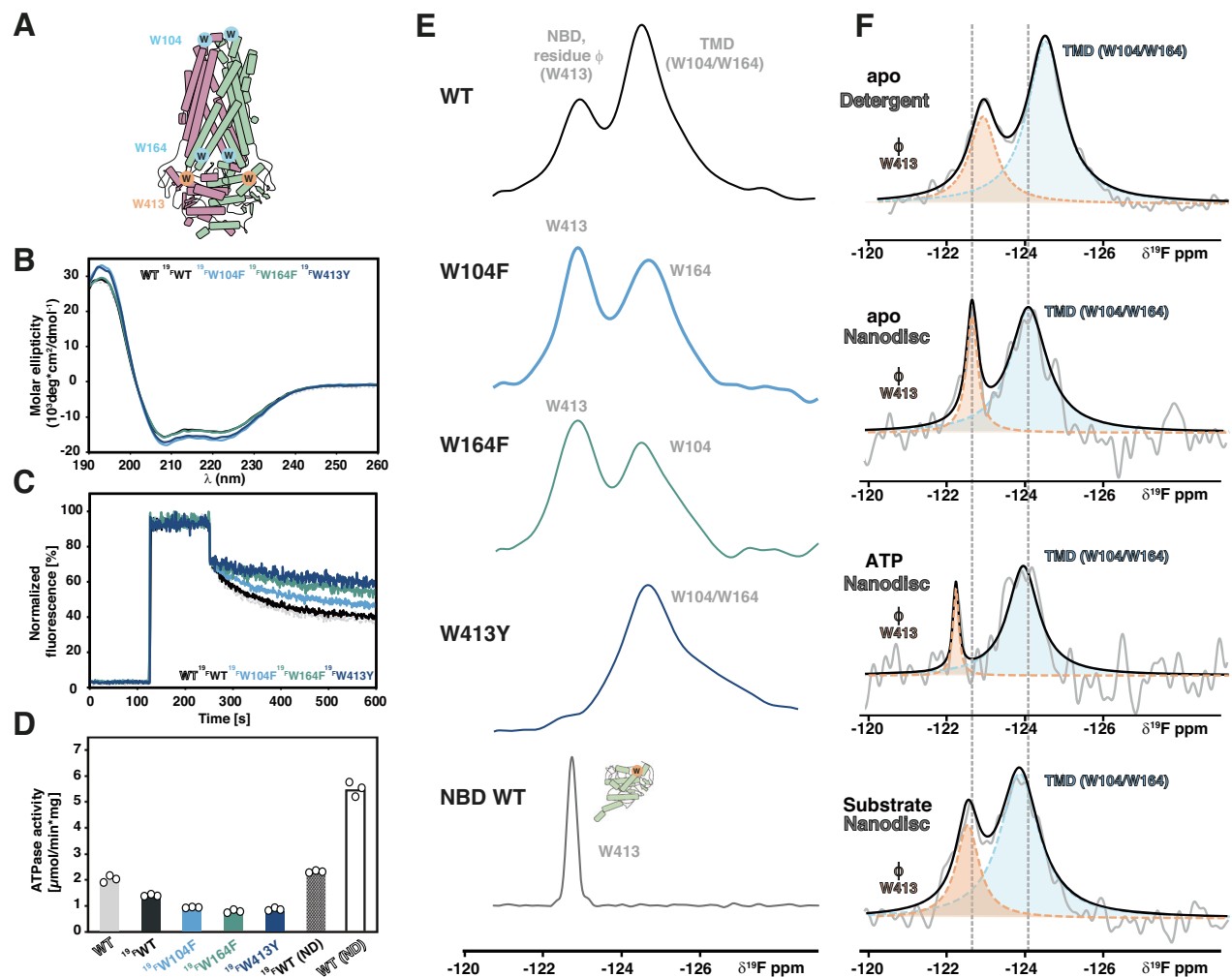

**Fig. 5 | The hinge senses both nucleotide and substrates. A** Position of the three native tryptophan residues in BmrA, W413 is the only tryptophan residue in the NBD and part of the hinge (residue φ, orange) (illustration based on PDB entry 6R81[29]). **B–D** Fluorination of tryptophan residues in full-length BmrA does not have a major impact on structural and functional integrity as seen by Circular dichroism spectroscopy, transport and ATPase assays. Fluorescence-based transport assay was carried out with doxorubicin in inside out vesicles prepared from cells over-expressing BmrA variants. All traces were normalized to the unlabeled WT curve (light grey). ATPase activity was measured in DDM/NaCholate (first five bars) or reconstituted in MSP1E3D1 nanodiscs prepared with *E. coli* polar lipid extract (last two bars on right, signified by 'ND'). Results shown are the mean of three technical triplicates. **E** 1D ¹⁹F NMR on detergent solubilized ¹⁹F-5Trp-labeled full-length BmrA. The three native tryptophan residues (W104, W164, W413) were individually mutated to obtain the respective ¹⁹F resonance assignments. For comparison, the spectrum of the isolated NBD with W413 is shown. **F** Nucleotide and substrate binding both are sensed by W413. ¹⁹F NMR assignments in the detergent state could be directly transferred to ¹⁹F-5Trp labeled BmrA reconstituted in MSP1E3D1 nano-discs with *E. coli* polar lipids. Addition of ATP or the substrate reserpine resulted in changes in chemical shift and linewidths for the ¹⁹F resonance of W413. For better visibility, we carried out a spectral deconvolution using a pure Lorentzian–line shape fitting.

communication hinge dynamics. In return, hinge mutations affect global protein dynamics and interdomain crosstalk.

## Discussion

ABC transporters are multidomain membrane proteins that rely on precise coordination between NBDs and TMDs for ATP-dependent substrate transport. We hypothesized that such coordination requires specific regions that sense both nucleotide and substrate binding. Using sequence alignments, functional assays, ¹H, ¹⁵N and ¹⁹F solution NMR, HDX-MS and PET-FCS on the bacterial type IV MDR transporter BmrA and its isolated NBD, we identified three highly conserved residues that influence transporter stability, ATPase and transport activities. These residues are structurally and dynamically coupled to each other as well as to distant regions of the transporter, forming a communication hinge at the NBD/TMD interface that links nucleotide and substrate binding sites via an L-shaped path-way (Fig. 6A).

Ligand binding or mutations at either end influence hinge dynamics, while hinge mutations alter ligand responses and activity, underscoring the bidirectional nature of this crosstalk. Mutations in the central hinge residue, W413, can uncouple ATP hydrolysis from substrate transport, additionally highlighting its critical role in the catalytic cycle.

These findings raise two key questions: What is the underlying mechanism by which signals such as nucleotide binding are relayed to the hinge? And is this a conserved feature in type IV ABC transporters, or possibly the whole ABC family?

In the apo state, the backbone amide of the Walker A residues show high dynamics and broadening (Supplementary Fig. 2B). Upon nucleotide binding, this region becomes more rigid, stabilizing the Walker A helix and likely influencing R398 (R^WA) at its C-terminus, thereby linking nucleotide sensing to hinge dynamics. Mutation of the Walker A motif (K380A) induces dynamic changes that propagate through the intervening R^WA residue to the hinge, consistent with the

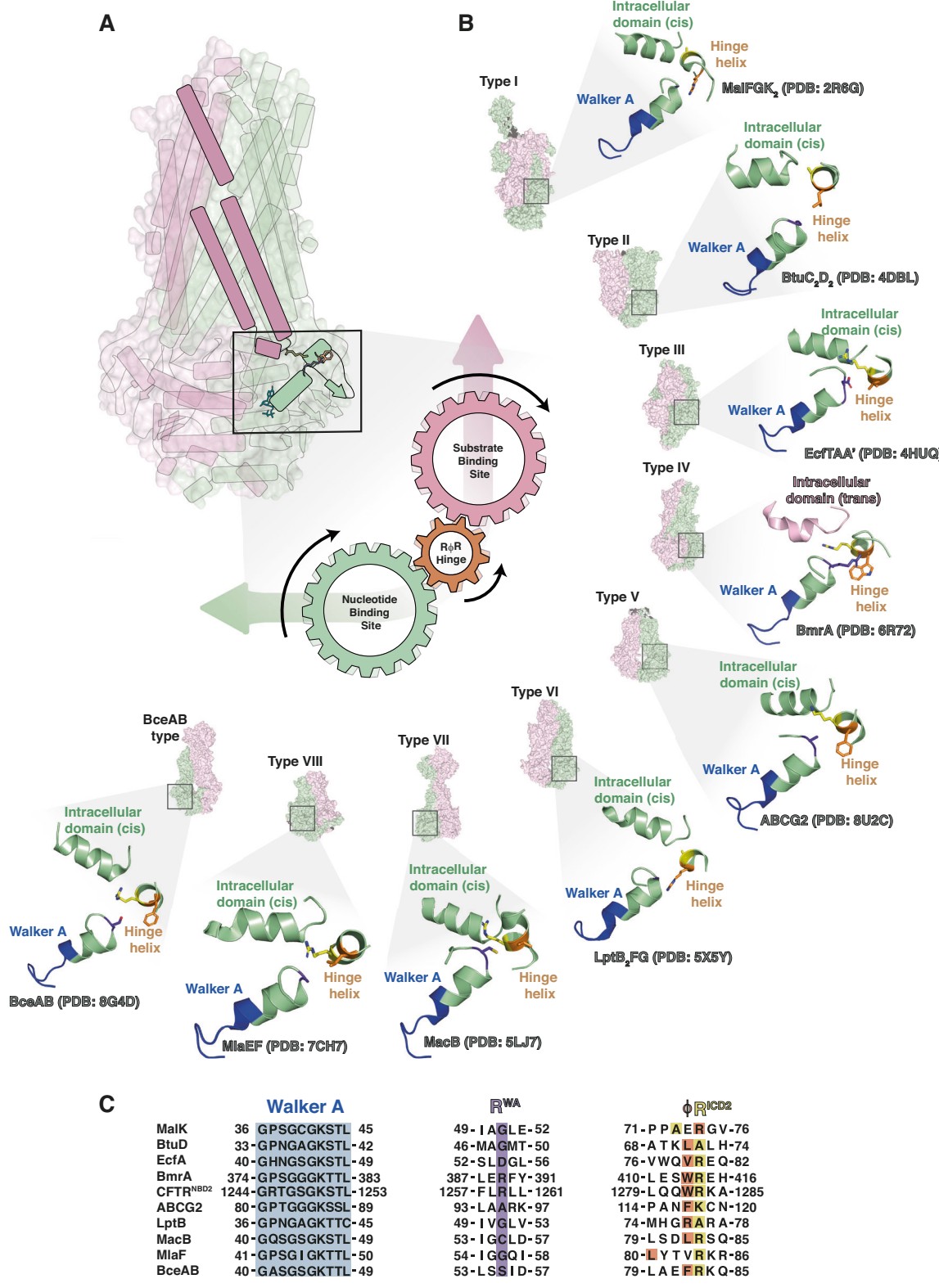

**Fig. 6 | Common features of the hinge region across ABC transporter families.**
**A** The hinge connects nucleotide and substrate binding site. Like cogwheels, the residues enable crosstalk via ICD2 (illustration based on PDB entry 6R81[29]).
**B** Comparison of the core elements of the hinge in different families of ABC transporters. Residues $R^{WA}$-φ – $R^{ICD2}$ are shown in orange, Walker A motif in blue and the intracellular domain in green (cis) or pink (trans). ABC transporter families I-VIII and BceA are presented by surface depictions of representative members and a zoom into the respective hinge region, i.e. MalFGK$_2$ (PDB: 2R6G[62]), BtuC$_2$D$_2$ (PDB:

4DBL[81]), EcfTAA' (PDB: 4HUQ[82]), BmrA (PDB: 6R81[29]), ABCG2 (PDB: 8U2C[83]), LptB$_2$FG (PDB: 5X5Y[84]), MacB (PDB: 5LJ7[85]), MlaEF (PDB: 7CH7[86]) and BceAB (PDB: 8G4D[87]). **C** Sequence alignment of the Walker A and hinge region for the ABC transporters shown in (**B**) was carried out using Clustal Omega[63]. Note that in some cases, the sequence alignment was adapted to reflect the 3D structure of the respective proteins (see main text for details). In LptB$_2$FG and MalFGK$_2$, an inversion of the properties of the sidechains in the positions φ and $R^{ICD}$ is seen, in MalFGK$_2$ and MlaEF residues φ and $R^{ICD}$ are not contiguous.

prevention of the transition from the inward- to the outward-facing state and its abolished ATPase activity. Notably, mutation of R414 produces a similar effect to the K380A mutation, supporting the model that conformational changes are transmitted through this L-shaped communication hinge.

In addition to local stabilization, large scale conformational changes and structural rigidification upon ligand binding are a hallmark of allosteric coupling in type IV transporters[12,25,50]. Both nucleotide binding and hinge mutations alter RecA/α-helical subdomain dynamics, as shown by PET-FCS, suggesting this step is critical for signal transmission between NBD and TMD. The groove between NBD subdomains, which harbors coupling helix 2, was identified by HDX-MS as the main hinge-responsive region (Fig. 4). Globally, BmrA becomes less dynamic upon substrate or ATP binding. It is thus conceivable that the hinge acts as "dynamic buffer", enabling the protein to interact with structurally diverse substrates, which may impose different structural demands on the NBD/TMD interface. Since hinge mutations can uncouple ATPase and transport function, it seems to be long-range communication, rather than inherent protein function, that is compromised when the hinge is disturbed.

In addition to substrate or nucleotide binding, the hinge may also contribute to lipid dependence of ABC transporter function[51]. Adding ABC transporters to the growing list of membrane proteins for which $^{19}F$ solution NMR spectroscopy has been successfully applied[35,52–54], we investigated detergent solubilized and nanodisc-reconstituted $^{19}F$-Trp-labeled BmrA (Fig. 5). Nucleotides, drugs and the membrane environment all affected the dynamics of hinge residue W413, suggesting that in addition to bidirectional intramolecular crosstalk, the hinge may also integrate exogenous cues such as membrane composition or fluidity.

Among the hinge residues, substitutions of residue φ were generally better tolerated than mutations of either of the two conserved arginine residues (Fig. 1). The unique ability of arginine to mediate numerous interactions, including hydrogen bonds, salt bridges, cation-π and π-π interactions between its guanidinium group and aromatics, as well as hydrophobic interactions likely play a decisive role[55,56]. While conservative substitutions of $R^{WA}$ retained at least partial activity, even a lysine substitution for $R^{ICD2}$ led to an inactive transporter. In structures of BmrA, the sidechain of $R^{ICD2}$ points towards ICD2 (Supplementary Fig. 14). The three hinge residues may thus act like interlocked cogwheels, connecting the nucleotide binding site to the hinge via the Walker A helix and residue $R^{WA}$, and the hinge to the substrate interaction sites and vice versa via residue $R^{ICD2}$ and the coupling helix 2 (Fig. 6A).

In line with studies on MsbA[15,17], a related type IV ABC transporter, our results show that while both ICD1 and ICD2 are crucial for driving the conformational changes of the catalytic cycle, hinge mutations primarily affect ICD2. Within the family of type IV transporters, the importance of the hinge identified here in BmrA may thus be of general relevance, despite some sequence variability (Fig. 6B, C). For instance, in CFTR (cystic fibrosis conductance regulator, ABCC7), NBD2 features a QWR sequence, while in NBD1, a glycine residue is found in position $R^{WA}$, and the WR residues in φ and $R^{ICD2}$ are present but not contiguous. Nonetheless, mutations causing cystic fibrosis are found in the hinge, including a deletion mutant adjacent to the residue homologous to BmrA R389 and at positions corresponding to W413 and R414[57]. Moreover, the side chain of residue F508 points towards the CFTR hinge helix. This residue is deleted in around 70% of all cystic fibrosis patient and results in a trafficking defect due to folding defects[58], however, small molecule correctors that stabilize the TMD can rescue these defects[59]. Similarly, our observation that some hinge mutants are misfolded in the context of the isolated BmrA NBD and only could be rescued in the presence of the TMD, suggest that a destabilized NBD may generally require a stable TMD platform to enable proper folding, emphasizing the importance of the coupling helices for transporter

stability. Furthermore, this highlights the value of studying isolated NBDs to derive structural and mechanistic details of ABC transporters, as they retain functional and structural integrity for nucleotide binding despite remaining monomeric[60].

While other ABC transporter classes lack the swapped domain topology of type IV transporters and the overall sequence conservation is low (Fig. 6C), some structural features persist throughout the superfamily. A short helix near the C-terminal tip of the Walker A helix typically harbors a hydrophobic residue (position φ) followed by a positively charged residue ($R^{ICD2}$). In some cases, such as in the type VI lipid transporter $LptB_2FG$[61] and the maltose transporter $MalFGK_2$[62], sidechain properties at positions φ and $R^{ICD}$ appear switched, raising the possibility that signal integration at this interface may be a general feature across the ABC superfamily.

In conclusion, we identified a previously uncharacterized, interdomain transmission pathway between NBD and TMD in a type IV ABC transporter that functions as a dynamic bidirectional relay integrating multiple signals to ensure proper ATP-dependent substrate transport. This not only positions the communication hinge as a central regulator of interdomain communication in an important ABC transporter but also provides a framework for studying similar relay elements in other complex biomacromolecular machines.

## Methods

### Reagents
All chemicals were purchased from Sigma-Aldrich, Roth and VWR unless otherwise stated. Reagents used include doxorubicin (Cayman Chemicals and Sigma), $^{15}N$-$NH_4Cl$, $^{13}C_6$-glucose (Eurisotop), 5-fluorotryptophane (BLD pharma) and ATTO oxa11 maleimide (ATTO-TEC). Lipids were purchased from Avanti Polar Lipids and Cayman Chemicals.

### Computational tools
Freely available computational tools were used to investigate the properties of BmrA constructs: Sequence alignment was carried out with Clustal Omega[63] (https://www.ebi.ac.uk/jdispatcher/msa/clustalo) and sequence logos created using WebLogo 3[64] (https://weblogo.threeplusone.com/). To create the WebLogo, 226 sequences were retrieved from the UniProtKB subdatabase (as of Nov. 27, 2024) which is part of the UniRef50_O06967 (50%) database. The latter corresponds to non-redundant protein sequences which show at least 50% sequence identity with BmrA. The sequences in the UniProtKB were either manually annotated and computationally analyzed (one entry, UniProtKB reviewed (Swiss-Prot)) or only computationally curated from (227 entries, UniProtKB Unreviewed (TrEMBL)) to remove notably protein fragments. One entry was still a protein fragment (A0A7Y8S210_BACSP) and a second one contains one residue insertion between the Walker A and the Q-loop motif (A0A398D0Y7_9BACL). These two sequences were removed from our database and the remaining 226 sequences were used to make the WebLogo shown in Fig. 1. NMR spectra were analyzed with Bruker TopSpin 4.0.8, CARA[65] and CCP NMR[66]. HDX-MS data were analyzed with PLGS™ software (ProteinLynx Global SERVER 3.0.2 from Waters™), DynamX 3.0 software (Waters™), and Deuteros 2.0 software[67].

### Cloning, expression, and purification NBDs
Synthetic genes coding for WT BmrA NBD (residues G331 – G589) from *Bacillus subtilis* and WT LmrA-NBD (residues D330 – Q590) from *Lactococcus lactis* cloned into pET-11a vector with a N-terminal His₆-tag followed by a TEV cleavage site were obtained from GenScript (Piscataway Township, NJ, USA). Gene coding for WT MsbA-NBD (residues 323 – 582) was amplified from *E. coli* BL21(DE3) gold (Agilent Technologies) and cloned into pET-11a vector with a N-terminal His₆-tag followed by a TEV cleavage site via Gibson assembly. Point mutations R389A, R389E, R389K, R389M, K380A, W413A, W413L, W413F, W413Y, R414A and R414K, C436S, N459W and S516C for BmrA-NBD, R397 and

W421 for LmrA NBD and R391A, L415A and L415W for MsbA NBD were introduced via site directed mutagenesis (see Supplementary Table 6 for primer sequences).

Transformed *E. coli* BL21 (DE3) Gold cells were growth at 37 °C in 1 L of Lysogeny Broth (LB) media supplemented with 100 μg/mL ampicillin until an $OD_{600}$ of 0.6 was reached. Immediately a final concentration of 1 mM IPTG was added and cells were grown at 21 °C overnight. Cells were harvested by centrifugation (5000xg, 15 min, 4 °C) and the resulting cell pellet was frozen in liquid nitrogen and stored at − 20 °C until further use. Purification followed our previously established protocol[28], but with an additional final concentration of 2.5 mM ADP added during lysis for R389A, R389E, R389K and R389M constructs to prevent protein aggregation.

### Labeling of NBDs for NMR and PET-FCS
Proteins were expressed and purified as described above with some modifications. $^1H$, $^{15}N$-labeled BmrA NBD WT, K380A, R389K, R389M, W413F, R414A and R414K, MsbA NBD WT and LmrA NBD WT for NMR spectroscopy were obtained by growing cells in M9 minimal medium[68] supplemented with $^{15}N$-$NH_4Cl$ as the sole nitrogen source. $^2H$,$^{13}C$, $^{15}N$-labeled MsbA NBD WT for backbone NMR assignments were obtained by growing cells in Silantes OD2 *E. coli* medium (Silantes GmbH, Munich, Germany). $^{19}F$-tryptophane-labeled BmrA NBD WT, K380A, R389K, R389M and R414K were obtained by growing cells in defined medium[68] supplemented with 50 mg/L 5-fluorotryptophane.

To obtain fluorescently modified NBD constructs for PET-FCS, Ni-NTA-column-bound single cysteine mutants were incubated for 1 h at room temperature with 10 mM TCEP. Afterwards, resin was washed with 10 column volumes (CV) washing buffer (50 mM Tris HCl pH 8, 500 mM NaCl) and incubated with 15-fold excess of ATTO oxa11 maleimide (ATTO-TEC) for 2 h at room temperature. To remove excess of fluorophore, the resin was washed with 10 CV of washing buffer (50 mM Tris HCl pH 8, 500 mM NaCl) and labeled protein was eluted and purified as previously described[28].

### Cloning, expression and purification of MSP1E3D1
Expression and purification of MSP1E3D1 was performed as previously described[69]. In brief, using the p1E3D1 plasmid (Addgene), MSP1E3D1 was expressed in *E. coli* BL21 cells. After harvest, bacteria were suspended in 50 mL of 40 mM Tris-HCl pH 7.4, 100 mM NaCl, 1 % (w/v) Triton X100, 0.5 mM EDTA, 1 mM PMSF. Two microliters of Benzonase (24 U/mL, Merck) were added and the bacteria were lysed with two passages at 18,000 psi through a microfluidizer 100 (Microfluidics IDEX Corp) and then centrifuged during 30 min. at 30,000 x g, 4 °C. The supernatant was loaded onto a 0.5-mL Ni$^{2+}$-NTA column (GE Healthcare) resin pre- equilibrated with 5 resin-volumes of 40 mM Tris-HCl pH 7.4, 100 mM NaCl, 1 % (w/v) Triton X100, 0.5 mM EDTA and 1 mM PMSF. The resin was then washed with 10 resin- volume with 3 different buffers: wash buffer 1 composed of 40 mM Tris-HCl pH 8.0, 300 mM NaCl and 1% (w/v) Triton X100; wash buffer 2 composed of 40 mM Tris-HCl pH 8.0, 300 mM NaCl, 50 mM sodium cholate and 20 mM Imidazole; wash buffer 3 composed of 40 mM Tris-HCl pH 8.0, 300 mM NaCl, 50 mM imidazole. MSP1E3D1 was eluted with 15 mL of 40 mM Tris-HCl pH 8.0, 300 mM NaCl and 500 mM Imidazole. TEV (2 mg/mL, with 1 mg TEV for 40 mg MSP1E3D1) was added to remove the His tag during dialysis (using a 12-14 kDa cutoff membrane) against 300 mL 40 mM Tris-HCl, pH 7.4, 100 mM NaCl and 0.5 mM EDTA for 3 h and then against 700 mL of the same buffer, overnight at 4 °C. After dialysis, 20 mM imidazole was added and the solution loaded on a 0.5 mL Ni$^{2+}$-NTA column equilibrated with 20 mM Tris-HCl pH 7.4 and 100 mM NaCl to remove uncleaved protein and the His-tagged TEV protease. MSP1E3D1 was concentrated spinning at 5,000 x g with a 10 kDa cutoff Amicon Ultra-15. The concentrated samples were frozen in liquid nitrogen and stored at -80 °C.

### Cloning and expression of full-length BmrA constructs
Vector pET23a (+) coding for BmrA full-length WT with a C-terminal His$_6$ tag was obtained as previously described[7]. Point mutations for R389A, R389E, R389K, R389M, W413A, W413L, W413F, W413Y, R414A, R414K and K380A were introduced via site directed mutagenesis (see Supplementary Table 6 for primer sequences). Transformed *E. coli* C41 (DE3) cells were growth at 37 °C in 1 L of 2YT media supplemented with 100 μg/mL ampicillin until an $OD_{600}$ of 0.6 was reached. $^{19}F$-tryptophane-labeled WT, W104F, W164F and W413Y BmrA full-length were obtained by growing cells in defined medium[68] supplemented with 50 mg/L 5-fluorotryptophane. For expression induction, a final concentration of 700 μM IPTG was added and cells were grown at 25 °C for 4 h. Afterwards, cells were harvested by centrifugation (5000 x g, 15 min, 4 °C) and the resulting cell pellet was frozen in liquid nitrogen and stored at − 20 °C until further use.

### Inside – Out Vesicle (IOV) preparation
For membrane isolation, the frozen cell pellet was resuspended in lysis buffer (50 mM Tris/HCl pH 8, 5 mM MgCl$_2$, 1 mM DTT) supplemented with 5 μg/mL of DNase I according to available protocols[7,70]. To disrupt the cells, the suspension was passed three times through a microfluidizer (maximator) at 18,000 psi. The lysate was centrifuged at 15,000 g for 30 min at 4 °C to remove cell debris and insoluble proteins. After centrifugation, the supernatant containing the membranes was ultracentrifuged at 150,000 g for 1.5 h at 4 °C to harvest cell membranes. The supernatant was discarded and the pellet containing membranes was resuspended in resuspension buffer (50 mM Tris/HCl pH 8, 1.5 mM EDTA). To reduce possible impurities in the membranes, a second ultracentrifugation step was performed at 150,000 x g for 1 h at 4 °C. The pellet from the second ultracentrifugation step was homogenized with a Dounce homogenizer in homogenization buffer (50 mM Tris/HCl pH 8, 1 mM EDTA, 300 mM sucrose) and aliquots were frozen in liquid nitrogen and stored at −80 °C until further use.

For further analysis, total membrane protein was quantified in the IOVs using the Colorimetric bicinchoninic Acid (BCA) assay (Thermo Fisher Scientific, Waltham, USA) where absorption at 562 nm was measured to determine the protein concentration.

### Purification of full-length BmrA constructs
Full-length BmrA wildtype or its mutants were purified by solubilizing the membrane proteins, and diluting IOVs containing the over-expressed transporters to 2 mg/mL of total membrane protein using solubilization buffer (50 mM Tris/HCl pH 8, 50 mM NaCl, 10 mM imidazole, 1 mM DTT, 10% (v/v) glycerol and 1% (v/v) n-dodecyl-ß–D–maltopyranoside (DDM))[30]. The solution was incubated for 1 h at 4 °C and 200 rpm and then ultracentrifuged at 150,000 xg for 1 h at 4 °C to remove insoluble proteins. The clear supernatant was incubated with NiNTA beads previously equilibrated with equilibration buffer (50 mM Tris/HCl pH 8, 50 mM NaCl, 10% (v/v) glycerol, 10 mM imidazole pH 8, 0.0675% (v/v) DDM and 0.04% (v/v) Na–Cholate) for 1.5 h at 4 °C and 200 rpm. Afterwards, the mixture was transferred to a gravity flow column and washed with 5 CV of washing buffer (Tris/HCl pH 8, 500 mM NaCl and 10% (v/v) glycerol) followed by 20 cv of buffer wash A (Tris/HCl pH 8, NaCl 50 mM, 10% (v/v) glycerol, 25 mM imidazole, 0.0675% (v/v) DDM and 0.04% (v/v) Na–Cholate). BmrA constructs were eluted with 6 CV of elution buffer (50 mM Tris/HCl, 50 mM NaCl, 10 % (v/v) glycerol, 250 mM Imidazole pH 8, 0.0675% (v/v) DDM and 0.04% (v/v) Na–Cholate). To reduce the imidazole concentration and to perform a buffer exchange, proteins were dialyzed against Hepes/KOH pH 8, 50 mM NaCl, 10% (v/v) glycerol, 0.035% (v/v) DDM and 0.03% (v/v) Na–Cholate overnight at 4 °C. Dialyzed proteins were then concentrated and loaded onto a size exclusion column (Superdex200 Increase 10/300 GL (Cytiva)) previously equilibrated with SEC buffer (Hepes/KOH pH 8, 50 mM NaCl, 10% (v/v) glycerol, 0.035% (v/v) DDM and 0.03% (v/v) Na–Cholate) via an NGC

chromatography system (Bio-Rad). The fractions containing pure protein were pooled and sample purity was verified by SDS-PAGE. Aliquots were frozen in liquid nitrogen and stored at $-80\,°C$ until further use.

## Reconstitution in liposomes

Proteoliposomes were prepared as described in Orelle et al., 2003[71]. Briefly, a stock of 25 mg/mL *E. coli* lipid extract was prepared in autoclaved MilliQ water. 10 µL of 10% DDM were added to 40 µL of the lipid stock in a 2 mL reaction tube at room temperature. After 1 h at constant stirring, 50 µg of previously purified transporter in DDM/Na-Cholate were added and the final volume adjusted to 250 µL with reconstitution buffer (50 mM Hepes/KOH pH 8 and 50 mM NaCl). After 1 h incubation at room temperature and constant stirring, three subsequent additions of 20 mg of dried Bio-Beads SM2 were done every hour. Afterwards, the mixture was centrifugated at 6000xg and supernatant containing proteoliposomes was kept at 4 °C until further use.

## Reconstitution in nanodiscs

Membrane Scaffold Protein (MSP) nanodiscs were prepared following the protocol by Alvarez et al.[69]. A stock of 25 mg/mL *E. coli* lipid extract was prepared in autoclaved MilliQ water. 9.3 µL of the lipid stock were mixed with 15 µL of 10% DDM in a 2 mL reaction tube at room temperature. After 1 h at constant stirring, 100 µg of purified transporter in DDM/Na-Cholate and 96 µg of purified MSP1E3D1 were added and the final volume adjusted to 250 µL with reconstitution buffer (50 mM Hepes/KOH pH 8 and 50 mM NaCl). After 1 h of incubation at room temperature and constant stirring, 170 mg of dried Bio-Beads SM2 were added and the mixture was incubated for 3 h at room temperature and constant stirring. To obtain the nanodisc reconstituted ABC transporter, the mixture was centrifuged at 6000 x g and the supernatant containing nanodiscs was kept at 4 °C until further use.

## Doxorubicin transport assay

Transport of doxorubicin by overexpressed BmrA transporters was studied in IOVs. The excitation and emission wavelengths were set at 480 and 590 nm, respectively. One hundred µg of IOVs containing BmrA wildtype or variants were diluted to 1 mL with transport buffer (50 mM Hepes/KOH pH 8, 8.5 mM NaCl, 4 mM phosphoenolpyruvate (PEP), 2 mM $MgCl_2$ and 60 µg of pyruvate kinase) in a 10 mm quartz cuvette (Hellma Analytics, Müllheim) and placed in a fluorimeter (Horiba or Photon Technology International, Inc.). Measurements were carried out at 25 °C under constant stirring with spectral bandwidths of 2 and 4 nm for excitation and emission, respectively. Fluorescence was recorded for 125 s before 10 µM of doxorubicin was added and the fluorescence was recorded for an additional 125 s. After these 250 s, ATP was added to a final concentration of 2 mM and the measurement continued for another 350 s.

## ATPase activity

The ATPase activity of detergent-purified or reconstituted BmrA constructs was determined based on an ATP-regenerating system coupling ATP hydrolysis to NADH oxidation which can be monitored spectrophotometrically at 340 nm. Using previously published protocols with slight modifications[7,70], NADH oxidation was monitored by measuring loss of absorbance at 340 nm in an UV-Vis spectrophotometer (SAFAS SP2000, Monaco) at 37 °C over 15 min. For each measurement, 3 µg of BmrA in detergent micelles or 1 µg of reconstituted transporter in nanodiscs or liposomes were diluted to 700 µL with ATPase buffer (50 mM Hepes/KOH pH 8, 10 mM $MgCl_2$, 4 mM phosphoenolpyruvate, 32 µg/mL lactate dehydrogenase, 60 µg/mL pyruvate kinase and 0.3 mM NADH) in a 10 mm quartz cuvette (Hellma Analytics, Müllheim). For the protein in detergent micelles, DDM and Na-Cholate were added to final concentrations of 0.035% (v/v) and 0.03% (v/v), respectively.

## Analytical size-exclusion chromatography (SEC)

20 µM of purified ABC transporter constructs, either isolated NBDs in 50 mM BisTris pH 7, 50 mM NaCl or full-length constructs in 50 mM Hepes/KOH pH 8, 50 mM NaCl, 10% glycerol, 0.035% DDM and 0.03% Na-Cholate were used. Samples were injected on a Superdex200 Increase 10/300 GL (Cytiva) column via an NGC chromatography system (Bio-Rad).

## Circular dichroism (CD) spectroscopy

CD measurements were conducted on a Jasco J-1500 CD spectrometer (Jasco, Gross-Umstadt, Germany) with 1 mm quartz cuvettes. For isolated NBD constructs, 5 µM protein in 5 mM Tris pH 7 and 2.5 mM NaCl were used. For the whole transporters, 1 µM protein was used in 0.5 mM Hepes/KOH pH 8, 1.5 mM NaCl, 0.035% DDM and 0.03% Na-Cholate. Spectra were recorded at 25 °C in a spectral range between 190 and 260 nm with 1 nm scanning intervals, 1.00 nm bandwidth and 50 nm/min scanning speed. All spectra were obtained from the automatic averaging of five measurements.

## Thermal stability assays

Nano differential scanning fluorimetry using the Prometheus NT.48 instrument (Nanotemper technologies, DE) was used[30,72]. BmrA samples at 1 mg/mL were incubated for 15 min at room temperature in the absence of ligands or in the presence of 10 mM ATP, 10 mM $MgCl_2$ and 1 mM sodium orthovanadate ($V_i$). 10 µL of the sample was used to fill the capillaries. A temperature gradient of 1 °C/min from 20 to 95 °C was applied and fluorescence was recorded at 330 and 350 nm. The ratio of fluorescence intensities at 350/330 nm was used to determine the melting temperature ($T_m$).

In addition, the melting temperatures ($T_m$) of the protein constructs were determined with a fluorescence-based assay[73]. Purified protein (5–25 µg) in 50 mM BisTris pH 7, 50 mM NaCl were used. Proteins were measured in the absence of ligands or incubated with 10 mM ADP or ATP with and without 10 mM $MgCl_2$. 2.5 µL of a 50X SYPRO Orange dye stock was added to each sample directly before measuring the melting temperature in a 96-well plate. Measurements were carried out on a QuantumStudio 1 Real-Time PCR System reader (Thermo Fisher) with a temperature increase of 0.05 /min. The fluorescence of SYPRO Orange was measured using for a SYBR GREEN-calibrated filter with an excitation filter of 470 +/– 15 nm and an emission of 520 +/– 15 nm.

For ABC transporters WT, K/A and R389X mutants, experiments were carried out with 10–25 µg of purified protein in Hepes/KOH pH 8, 50 mM NaCl, 10% (v/v) glycerol, 0.035% (v/v) DDM and 0.03% (v/v) Na-Cholate. The samples were measured in the absence of ligands or incubated with 10 mM ADP or ATP with 10 mM $MgCl_2$. 2 µL of a 10X GloMelt fluorescent dye stock was added to each sample directly before measurement in a 96-well plate. The melting temperatures were determined with a QuantumStudio 1 Real-Time PCR System reader (Thermo Fisher) with a temperature increase of 0.05 °C /min. The fluorescence of GloMelt was measured using the filter calibrated for SYBR GREEN with an excitation filter of 470 +/– 15 nm and an emission of 520 +/– 15 nm.

## NMR Spectroscopy

For protein backbone assignments of $^2H,^{13}C,^{15}N$-labeled MsbA NBD WT, sample was concentrated to 450 µM in 50 mM BisTris pH 7, 50 mM NaCl added with 10 mM ADP, 10% v/v $D_2O$ and 0.15 mM DSS (final concentrations). TROSY-based $^{15}N$-HSQC, HNCA, HNCO, HN(CA)CO, HN(CO)CA, HNCACB and HNCOCACB experiments were recorded on a Bruker AVANCE 600 MHz spectrometer equipped with cryogenic triple resonance probe (Bruker GmbH, Karlsruhe, Germany) at 298 K using standard NMR pulse sequences implemented in the Bruker Topspin library. Backbone NH assignments were validated using MsbA NBD WT labeled with $^{15}N$-lysine, arginine, tyrosine, phenylalanine,

valine, leucine, isoleucine or serine/glycine. Spectra were processed using Bruker TOPSPIN 4.0.8 and analyzed using CARA[65]. Backbone assignments of BmrA[28] and LmrA[27] have been reported by us previously.

For chemical shift perturbation assays, samples were concentrated to 200 – 400 µM before addition of 10 mM ADP or ATP and 10% v/v $D_2O$ and 0.15 mM DSS (final concentrations). TROSY – based $^1H$, $^{15}N$-HSQC NMR spectra of isotope labeled NBDs in 50 mM BisTris pH 7, 50 mM NaCl were recorde ord at 298 K on a Bruker AVANCE 600 or Bruker Neo 800 MHz spectrometer equipped with cryogenic triple resonance probes (Bruker GmbH, Karlsruhe, Germany) using standard NMR pulse sequenced implemented in Bruker Topspin library. All spectra were processed using Bruker TOPSPIN 4.0.8 and analyzed using CCP NMR v2.5 analysis[66].

The average $^1H$ and $^{15}N$ weighted chemical shift perturbations (CSP) observed in $^1H$, $^{15}N$ HSQC spectra can be calculated according to Eq. 1[74]:

$$CSP = \sqrt{0.5*[\triangle\delta_H^2 + (0.15*\triangle\delta_N)^2]} \qquad (1)$$

Here, $\triangle\delta_H$ is the $^1H$ chemical shift difference, $\triangle\delta_N$ is the $^{15}N$ chemical shift difference, and CSP is the average $^1H$ and $^{15}N$ weighted chemical shift difference in ppm.

## $^{19}F$ NMR

$^{19}F$-NMR spectra of 5-fluorotryptophane labeled BmrA NBD and full-length constructs were recorded at 298 K on a Bruker AVANCE III 600 MHz spectrometer equipped with a Prodigy TCI or a CP2.1 QCI cryoprobe (Bruker GmbH, Karlsruhe, Germany) using standard NMR pulse sequences implemented in Bruker Topspin library. For isolated NBDs all measurements were carried out in 50 mM BisTris pH 7, 50 mM NaCl. Samples were concentrated to 100 – 250 µM before addition of 10 mM ADP and 10% v/v $D_2O$ and 0.15 mM DSS (final concentrations). In the case of 5-fluorotryptophane labeled BmrA full-length constructs, all measurements were carried out in Hepes/KOH pH 8, 50 mM NaCl, 10% (v/v) glycerol, 0.035% (v/v) DDM and 0.03% (v/v) Na–Cholate. Samples were concentrated to 100 – 300 µM before addition of 10 mM ADP, 10 mM ATP or 15 mM reserpine and 10% v/v $D_2O$ and 0.15 mM DSS (final concentrations). All spectra were processed using Bruker TOPSPIN 4.0.8 and deconvoluted using pure Lorenztian-line-shape fitting.

## PET-FCS experiments

Photoinduced electron transfer fluorescence correlation spectroscopy (PET-FCS) was performed using a custom-built confocal fluorescence microscope setup described elsewhere[75]. Fluorescently modified NBD constructs were diluted to 1 nM final concentration in 50 mM phosphate buffer pH 7.0 with the solution ionic strength adjusted to 200 mM using potassium chloride. 0.3 mg/ml Protease-free bovine serum albumin and 0.05% Tween-20 were applied as solution additives to suppress sample/glass-surface interactions. The buffer was filtered through a 0.2 µm syringe filter. To study the effect of nucleotides, 1 nM protein samples were incubated with 10 mM ADP or ATP directly prior to measurement and then transferred onto a microscope slide and covered by a cover slip. The sample temperature was adjusted to 25 °C using a custom-built objective heater. The total measurement time was 15 min for each sample.

## PET-FCS data analysis

Autocorrelation functions were analyzed by fitting a model for a two-dimensional diffusion of a globule containing a sum of $n$ single-exponential relaxations[39,76]:

$$G(\tau) = \frac{1}{N}\left(1 + \frac{\tau}{\tau_D}\right)^{-1}\left(1 + \sum_n a_n \exp\left(-\frac{\tau}{\tau_n}\right)\right) \qquad (2)$$

$t$ is the lag time, $N$ is the average number of molecules in the detection volume, $\tau_D$ is the experimental diffusion time constant, $a_n$ and $\tau_v$ are the observed amplitude and time constant of the $n^{th}$ relaxation. The application of a model for diffusion in two dimensions was of sufficient accuracy because the two horizontal dimensions $(x, y)$ of the detection focus were much smaller than the lateral dimension $(z)$ in the applied setup.

## HDX-MS

Hydrogen deuterium exchange coupled to mass spectrometry (HDX-MS) experiments were performed using a Synapt G2-Si mass spectrometer coupled to a NanoAcquity UPLC M-Class System with HDX Technology (Waters™). All the reactions were carried out manually, as previously described[29]. Deuteration labeling was initiated by diluting 5 µL of 20 µM BmrA W413F or 5 µL of 15 µM BmrA WT reconstituted in nanodiscs in 95 µL $D_2O$ labeling buffer (5 mM Hepes pD 8.0, 50 mM NaCl). For ADP/Vi trapping, the labeling buffer additionally contained 10 mM ATP, 10 mM $MgCl_2$, 1 mM Vi. For the doxorubicin-bound condition, the labeling buffer additionally contained 100 µM doxorubicin. Prior to labeling, the samples were incubated with the respective ligands for 15 min at 20 °C. Samples were labeled for 2, 5, 15 and 30 min at 20 °C. Subsequently, the reactions were quenched by adding 22 µL of ice-cold quenching buffer (0.5 M glycine, 8 M guanidine-HCl pH 2.2, 0.035% DDM and 0.03% sodium cholate) to 100 µL of labeled sample, in ice bath. After 1 min, the 122 µL quenched sample was added into a microtube containing 200 µg of activated zirconium magnetic beads (MagRe-Syn Zr-IMAC from Resyn Biosciences, USA) to remove the phospholipids[77]. After 1 min, magnetic beads were removed, and the supernatant was injected immediately into a 100 µL loop. Labeled proteins were then subjected to on-line digestion at 15 °C using a pepsin column (Waters Enzymate™ BEH Pepsin Column 300 Å, 5 µm, 2.1 x 30 mm). The resulting peptides were trapped and desalted for 3 min on a C4 pre-column (Waters ACQUITY UPLC Protein BEH C4 VanGuard pre-column 300 Å, 1.7 µm, 2.1 x 5 mm, 10 K - 500 K) before separating them with a C4 column (Waters ACQUITY UPLC Protein BEH C4 Column 300 Å, 1.7 µm, 1 x 100 mm) using 0.2% formic acid and a 5–40 % linear acetonitrile gradient in 15 min and then 4 alternative cycles of 5% and 95% until 25 min. The valve position was adjusted to divert the sample after 14 min of each run from C4 column to waste, to avoid a contamination of the mass spectrometer with detergent. At least two full kinetics from the same protein sample were measured subsequently. Blanks (equilibration buffer: 5 mM Hepes pH 8.0, 50 mM NaCl) were injected after each sample injection and pepsin column washed during each run with pepsin wash (1.5 M guanidine-HCl, 4% acetonitrile, 0.8% formic acid pH 2.5) to minimize the carryover. Electrospray ionization mass spectra were acquired in positive mode in the m/z range of 50 – 2000 and with a scan time of 0.3 s. For the identification of non-deuterated peptides, data was collected in MSE mode and the resulting peptides were identified using PLGS™ software (ProteinLynx Global SERVER 3.0.2 from Waters™). Peptides were then filtered in DynamX 3.0 software (Waters™), with the following parameters: minimum intensity of 1000, minimum products per amino acid of 0.3 and file threshold of 2 out of 8–11. After manual curation, Deuteros 2.0 software was used for data analysis, visualization and statistical treatments for differential HDX-MS. The significance of identified changes of deuteration was evaluated using Peptide-level significance statistical tests[67] with a $p$-value threshold of 0.05. Unsignificant peptides were then manually removed in DynamX 3.0 software (Waters™). Data were finally integrated in PDB files using HDX-viewer tool[78] for 3D visualization. Figures were prepared using the PyMOL Molecular Graphics System (Version 3.0 Schrödinger, LLC[79]). The mass spectrometry data have been deposited to the ProteomeXchange Consortium via the PRIDE partner repository with the dataset identifier PXD027447 for BmrA

WT in the apo and ADP*Vi states and PXD058013 for the W413F mutant in the respective states as well as WT and mutant bound to doxorubicin.

## Reporting summary

Further information on research design is available in the Nature Portfolio Reporting Summary linked to this article.

## Data availability

The NMR data generated in this study have been deposited in the BioMagResBank (www.bmrb.io) database under accession code 52626. The HDX-MS data generated in this study have been deposited to the ProteomeXchange Consortium via the PRIDE[80] partner repository with the dataset identifier PXD058013 and PXD027447. The functional data generated in this study are provided in the Supplementary Information and Source Data file. The NMR data used in this study are available in the BioMagResBank (www.bmrb.io) database under accession code 51156 and 17660. The cryo-EM data used in this study are available in the Protein Data Bank (https://www.rcsb.org/) under accession codes 2R6G, 4DBL, 4HUQ, 5LJ7, 5X5Y, 6R81, 7CH7, 7OW8, 8G4D and 8U2C. Source Data are provided as a Source Data file. Source data are provided with this paper.

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

## Acknowledgements

CO and UAH are grateful to Amy L. Davidson for introducing us and for kindly providing feedback and continuous support. We thank Dres Elise Kaplan, Mai Anh Tran, Jan Overbeck and Christoph Wiedemann for fruitful discussions and technical support. VHPC acknowledges a DAAD-CONACYT PhD fellowship and a PROCOPE mobility 2020 fellowship by the Department for Science and Technology of the Embassy of France in Germany. DRS acknowledges a PhD fellowship and a research stay fellowship by the Hans Böckler Stiftung. We thank the Jena School for Microbial Communication (JSMC) at the Friedrich Schiller University Jena for support. This work was supported by the Deutsche Forschungsgemeinschaft, Project ID 421388231 (to HN and UAH), and the Cluster of Excellence "Balance of the Microverse" (EXC 2051—Project-ID 390713860) (to UAH) as well as by the Agence Nationale de la Recherche grant N° ANR-19-CE11-0023-01 (to CO) including a PhD fellowship for MDC. MDC is grateful to the Fondation pour la Recherche Médicale (FRM) for supporting a 6-months PhD extension (Funding number FDT202204015047). UAH acknowledges an instrumentation grant for a high-field NMR spectrometer by the REACT-EU EFRE Thuringia (Recovery assistance for cohesion and the territories of Europe, ERDF, Thuringia) initiative of the European Union. CO and J-MJ acknowledge the Protein Science Facility of SFR Biosciences (Universite Claude Bernard Lyon 1, CNRS UAR3444, Inserm US8, ENS de Lyon) for the use of the circular dichroism equipment.

## Author contributions

Sample preparation and biochemistry: VHPC, MDC; NMR spectroscopy: VHPC; Mass spectrometry: MDC, JM; Data analysis: all authors; Conceptualization: CO, JMJ, UAH; Funding acquisition JM, HN, CO, JMJ, UAH; Supervision: JM, CO, JMJ, UAH; Paper writing – first draft: VHPC, CO, JMJ, UAH; Paper writing—review and editing: MDC, DRS, WJ, HN, JM, CO, JMJ, UAH; visualization: all authors. All authors read and approved the final version of the manuscript.

## Funding

## Competing interests
