## [Transparent Peer Review file · Nature Communications]

Bidirectional communication between nucleotide and substrate binding sites in a type IV multidrug ABC transporter

Corresponding Author: Professor Ute Hellmich

Version 0:

Reviewer comments:

Reviewer #1

(Remarks to the Author)

In this paper, Carrillo, Cesare and colleagues investigate the molecular mechanism of type IV ABC transporters, focusing mostly on the prototypical BmrA multidrug efflux pump. Specifically, they examine how communication between the TMDs and NBDs occurs at the molecular level. The authors focus on a cluster of three residues located in ICD2 and demonstrate, using an impressive array of biophysical and biochemical approaches, that this cluster is important for ATPase activity and transport function. More importantly, they conduct an extensive characterization of the local and global dynamics of both the NBDs and the full-length protein, observing long-range crosstalk occurring at the molecular level upon nucleotide and substrate binding.

Their main finding is that there is allosteric coupling between this identified “hinge region” and the aforementioned domains (TMDs and nucleotide-binding sites). Nucleotide binding structurally influences the hinge region, as shown by PET-FCS, NMR, and HDX-MS. Reciprocally, mutations in this hinge region structurally affect the nucleotide-binding site within the NBDs of three different type IV transporters. Similarly, mutations in the hinge region notably diminish the structural effect of substrate and/or nucleotide binding to the full-length transporters, highlighting its role in relaying structural information between the TMDs and NBDs at the molecular level.

This work gathers an impressive amount of high-quality biophysical data, and as far as I can tell, I cannot provide any feedback regarding the experiments besides “well done.” The quality of the experimental data is excellent, and the accumulation of these results has certainly required an impressive amount of benchwork.

My major concern regarding this work is that the rationale for focusing on these three residues is not clear to me. My understanding is that a combination of spurious observations on the NBD of homologs, coupled with visual inspection of structures and previous studies pinpointing the ICD as a potential coupling element between the substrate-binding pocket and the NBD, played a role in their selection. However, why these three specific residues were chosen, and in particular the “bulky Trp/Leu” one, is still unclear to me after reading the manuscript. In this regard, Fig. 1B adds to the confusion, as the alignment of type IV ABC transporters shows other residues with a similar level of conservation (F415, for example) that do not receive the same attention. This is even more striking in the conservation plot of BmrA-like transporters, where many other residues appear to be equally conserved.

So, I would say that my general issue with this work is that, while I do not doubt that the identified residues are important, I can't help but wonder whether conducting similar experiments on other mutants in a comparable region—sandwiched between the NBD and TMD—would have yielded different results. I also acknowledge that in science, serendipity plays an important role and that not every experiment has to be strictly hypothesis-driven. However, clarification of the rationale behind focusing on these three residues, as opposed to other potential sites, would be beneficial.

Minor comments:

1. The abstract is a bit misleading. “Combining nuclear magnetic resonance (NMR) spectroscopy, Hydrogen-Deuterium eXchange Mass Spectrometry (HDX-MS), photo-induced electron-transfer fluorescence correlation spectroscopy (PET-FCS) and functional assays, we identified a conserved cluster of residues at the NBD/TMD interface of the bacterial MDR transporter BmrA.” It's rather the opposite – the cluster was identified using MSAs, and visual inspection of the structures etc. The functional and structural impact was validated using a combination of nuclear magnetic resonance (NMR) spectroscopy, Hydrogen-Deuterium eXchange Mass Spectrometry (HDX-MS), photo-induced electron-transfer fluorescence correlation spectroscopy (PET-FCS) and functional assays”.

2. In the introduction, a figure would be helpful, even in the supplementary, to get familiar with the ABC transporter nomenclature and what it corresponds to structurally (ID1, X-loop, RecA domain). Otherwise, reading of the following paragraph is arduous for a non ABC-transporter expert: “In this subfamily, interdomain crosstalk between NBD and TMD is mainly attributed to the proximity of the conserved Q-9,10 and X-loop motifs 11–13 of the NBDs with the coupling helices 14–17 of the TMDs. In half transporters like BmrA with six transmembrane helices, these intracellular α -helical linkers connect transmembrane helices 2 and 3 as well as 4 and 5. Together with the cytosolic regions of the respective transmembrane helices these linkers constitute the intracellular domains 1 and 2, i.e. ICD1 and ICD2, respectively¹⁸. Upon assembly into functional transporter dimers, ICD1 interacts with the NBD within the same subunit (in cis), while ICD2 reaches over to the NBD of the opposing subunit (in trans), inserting into a groove on the NBD surface between the RecA and the α -helical subdomains^{12,19}”.

3. I'm sorry to say that I don't see how PET FCS adds anything to the argument. NMR is less perturbative and so is HDX-MS. Could the authors clarify what is the additional insight brought by this method?

4. Why is there no direct comparison between WT and hinge mutants using HDX-MS? In general, the global view presented of the HDX data does not clearly show whether it supports the statement that “hinge mutations affect nucleotide and substrate binding” and that “nucleotide and substrate binding affect the hinge region.” Could a figure highlighting representative peptides from the nucleotide-binding region and the substrate-binding regions be shown for WT vs. the W139F mutant? Similarly, could peptides from the hinge region be shown for the apo, doxorubicin-bound, and ADP vanadate-bound states?

5. Maybe the authors should agree on a consistent terminology to describe the cluster of residues and its effects and stick to it throughout the paper. Terms like nexus, region, network, cluster—as well as long-range, allosteric, bi-directional, coupling, dynamic, etc.—are used interchangeably, leading to sentences like “with a long-range dynamic coupling network” that feel quite abstract.

6. There is a mistake in the color-coding of the Δ HDX data in Supplementary Figure 9: “more accessible and more protected” colors (red and blue) have been reversed compared to figure 3.

Reviewer #2

(Remarks to the Author)

This manuscript describes a critical role for 3 residues that act as a sort of hinge between the NBD and TMD of a Type IV ABC transporter, facilitating communication between the domains. This interdomain communication is critical for coupling of ATP hydrolysis and substrate transport. The data presented is primarily for BmrA, but sequence conservation and some chemical shift perturbation data on the MsbA and LmrA NBDs suggest that this hinge may be functionally important in other Type IV ABC transporters. This is a dense paper with a wealth of data. The authors make a convincing case for the importance of this 3-residue hinge, and the potential role of this novel hinge in regulating coupling within the Type IV ABC transporters makes this of interest to a broad community. The manuscript also demonstrates the value of evaluating mutations and the role of individual residues in both full length protein and isolated domains, as well as the importance of integrating an array of experimental methods to characterize such complex regulation of protein function. However, the range of methods, mutants and types of information provided by the different experiments makes the paper very complex. Careful presentation of the data and explanation in the text to focus on the key results and distinguish primary data and conclusions from supporting data sets is necessary to make the manuscript comprehensible to the broad readership of Nature Communications.

My primary concern is that the main message gets lost at times in the large amount of data. This is particularly true when the figures are so small that they are impossible to read or see in a printed copy. The color coding is very helpful, but sometimes light-colored fonts over colored backgrounds are difficult to read. Enlarging the individual panels to make sure the key features of the data is critical. It would also be helpful to divide some of the figures to clearly delineate data on the isolated NBD versus the full-length transporter, or to distinguish different types of data reporting on distinct regions and timescales so that each point is made clearly.

While the authors discuss the significant chemical shift perturbations (Fig. 1) or changes in peak position and lineshape (Fig. 2), there are some sites that have minimal chemical shift perturbation (Fig.1 and 2D RICD2, Fig. C iv). This is not a major concern in the context of the vast amount of data, but should be properly acknowledged. Particularly since RICD2 has minimal CSP in several experiments.

Reviewer #3

(Remarks to the Author)

The authors investigated a possible allosteric transmission pathway in ABC transporters. They first identified three conserved residues (R389, W413, R414 in BmrA) that appear to connect the nucleotide binding site in the Walker A motif to intracellular domain 2 (ICD2), providing a potential bridge to the substrate binding site. In a second step, the authors used NMR (HSQC) spectroscopy to understand whether the resonances of the three residues respond to the addition of

nucleotides, which they did, confirming a direct link between the Walker A motif and this hinge region. Using the full-length transporter, the authors found that drastic mutations of these residues severely impaired ATPase and transport activity. Interestingly, a W413Y/F mutation still showed significant ATPase activity but a severe loss of transport activity, suggesting that this mutation decouples the two tasks. They also showed that a mutation in one of the residues affects the NMR resonances of the others. In particular, mutations in the hinge also affect the resonances of K380, which is located in the Walker A motif. Using photo-induced electron transfer (PET) in combination with fluorescence correlation spectroscopy (FCS), the authors found that replacing W413 with a less bulky residue (F) drastically reduces the quenching amplitude, indicating a more rigid structure. Similar results were obtained using hydrogen-deuterium exchange (HDX) coupled to mass spectrometry, which also showed rigidification in the presence of a nucleotide transition state analogue. Finally, the authors investigated the effect of substrate-binding on the ¹⁹F-NMR resonances of fluorinated W413 in nanodiscs and concluded that substrate-binding also affects the line-width of this resonance, indicating an effect on the dynamics. Overall, this is an interesting study that uses an impressive set of tools to elucidate the allosteric coupling between substrate and nucleotide binding in ABC transporters. Overall, I am positive, but I have a number of questions that the authors should address in a revised version of the manuscript. In particular, the authors don't make it easy to follow their argument throughout, partly because it remains very descriptive (as opposed to mechanistic) and partly because some terms are not properly introduced.

1.) The fact that a mutation of K380 affects residues in the hinge could be explained by a change in the dipole of the helix. Could this be a way of establishing communication between the nucleotide binding site and the hinge? The authors remain rather vague about the actual mechanism of communication.

2.) Since the authors don't say, do the NBDs dimerise in exactly the same structure as the full-length dimer (in BmrA)?

3.) I had problems with what to make of the timescales in the PET-FCS experiment. It seems to me that these timescales are extremely fit sensitive (looking at the noise in the correlation functions). However, what is very clear from these experiments is that the amplitude drops significantly when W413 is replaced by phenylalanine.

4.) Related to the above point, I don't understand what is meant by three modes of action. Do the authors mean three structurally distinct states that interconvert on nano- to microsecond timescales? I think it is the lack of a picture of a potential mechanism that leads to the rather descriptive style of the paper. Perhaps the authors could be clearer about what they actually mean by modes of motion.

5.) More information is needed for the HDX experiments. Which exchange regime was used, EX1 or EX2? Can the deuterated fractions be compared between wt and phi mutant or can we only compare the differences between apo and holo?

6.) On page 7, lines 181 and 186, the RWA residue is referred to as R398, but it appears that R389 is correct.

7.) On page 6, line 194, the authors refer to an outward-facing state that hasn't been introduced before, causing some confusion. Perhaps this nomenclature is clear to anyone working on ABC transporters. It wasn't to me. I had to search the manuscript until I found Supplementary Figure 10, which explains that the transporter was crystallised in two forms. The authors would like to introduce this conformational flexibility much earlier, ideally in the introduction, as the interpretation of some of the results depends on it.

8.) Related to this point, the authors never specify which conformation or mixture of conformations the full-length transporter is in and how this equilibrium is controlled (is it controlled?). For example, in Figure 3A I had to conclude that the equilibrium is controlled by the nucleotide analogue.

9.) On p. 7, line 223, the authors state that Fig. 2B shows HSQC spectra. However, Figure 2B shows ¹⁹F NMR spectra.

Version 1:

Reviewer comments:

Reviewer #1

(Remarks to the Author)

The authors have satisfactorily addressed the concerns, and the manuscript is much improved. I have no objections to its publication.

Reviewer #3

(Remarks to the Author)

The authors addressed all my comments satisfactorily.

Reviewer #1 (Remarks to the Author):

In this paper, Carrillo, Cesare and colleagues investigate the molecular mechanism of type IV ABC transporters, focusing mostly on the prototypical BmrA multidrug efflux pump. Specifically, they examine how communication between the TMDs and NBDs occurs at the molecular level. The authors focus on a cluster of three residues located in ICD2 and demonstrate, using an impressive array of biophysical and biochemical approaches, that this cluster is important for ATPase activity and transport function. More importantly, they conduct an extensive characterization of the local and global dynamics of both the NBDs and the full-length protein, observing long-range crosstalk occurring at the molecular level upon nucleotide and substrate binding.

Their main finding is that there is allosteric coupling between this identified “hinge region” and the aforementioned domains (TMDs and nucleotide-binding sites). Nucleotide binding structurally influences the hinge region, as shown by PET-FCS, NMR, and HDX-MS. Reciprocally, mutations in this hinge region structurally affect the nucleotide-binding site within the NBDs of three different type IV transporters. Similarly, mutations in the hinge region notably diminish the structural effect of substrate and/or nucleotide binding to the full-length transporters, highlighting its role in relaying structural information between the TMDs and NBDs at the molecular level.

This work gathers an impressive amount of high-quality biophysical data, and as far as I can tell, I cannot provide any feedback regarding the experiments besides “well done.” The quality of the experimental data is excellent, and the accumulation of these results has certainly required an impressive amount of benchwork.

We sincerely thank the reviewer for this very positive and encouraging assessment.

My major concern regarding this work is that the rationale for focusing on these three residues is not clear to me. My understanding is that a combination of spurious observations on the NBD of homologs, coupled with visual inspection of structures and previous studies pinpointing the ICD as a potential coupling element between the substrate-binding pocket and the NBD, played a role in their selection. However, why these three specific residues were chosen, and in particular the “bulky Trp/Leu” one, is still unclear to me after reading the manuscript. In this regard, Fig. 1B adds to the confusion, as the alignment of type IV ABC transporters shows other residues with a similar level of conservation (F415, for example) that do not receive the same attention. This is even more striking in the conservation plot of BmrA-like transporters, where many other residues appear to be equally conserved. So, I would say that my general issue with this work is that, while I do not doubt that the identified residues are important, I can't help but wonder whether conducting similar experiments on other mutants in a comparable region—sandwiched between the NBD and TMD—would have yielded different results. I also acknowledge that in science, serendipity plays an important role and that not every experiment has to be strictly hypothesis-driven. However, clarification of the rationale behind focusing on these three residues, as opposed to other potential sites, would be beneficial.

We appreciate the opportunity to expand on the hypotheses based on informed by both previously published work (ours and others') and new data presented in this manuscript. Together, these findings have led us to conclude that the three residues we focus on represent the core components of a novel communication pathway between the NBD and TMD. We have thus extended the introduction and parts of the results and discussion section to highlight this more.

*In brief, our earlier solution NMR studies on the NBD of the *L. lactis* MDR transporter LmrA (PMID: 26449340) showed that the backbone and sidechain amide chemical shifts of a tryptophan residue remote from the nucleotide-binding site (W421) respond to nucleotide binding. This suggested that W421 may form part of a long-range nucleotide-sensing network. A corresponding tryptophan residue in the NBD of *B. subtilis* BmrA (W413) was previously shown by Trp fluorescence to be a sensitive reporter of drug binding (PMID: 15182191). Structurally, W413 is located in the NBD/TMD interface (see Fig. 1A main manuscript).*

These observations, together with our new data showing that W413 is also responsive to nucleotide binding and that its mutation can “uncouple” ATPase and transport activity, suggest that bacterial type IV MDR transporters harbor a previously unrecognized allosteric sensing region in the NBD/TMD interface for both nucleotides and substrates. Importantly, mutations at other sites in BmrA or related transporters did not produce the consistent functional phenotypes observed for the three hinge residues described here. While we do not rule out additional contributors to interdomain crosstalk, the effects of W413 mutations are particularly striking and consistent with a hinge-like role. Furthermore, W413 is directly affected by mutations in R398 and R414, and these residues greatly impact transporter function.

Minor comments:

1. The abstract is a bit misleading. “Combining nuclear magnetic resonance (NMR) spectroscopy, Hydrogen-Deuterium eXchange Mass Spectrometry (HDX-MS), photo-induced electron-transfer fluorescence correlation spectroscopy (PET-FCS) and functional assays, we identified a conserved cluster of residues at the NBD/TMD interface of the bacterial MDR transporter BmrA.” It’s rather the opposite – the cluster was identified using MSAs, and visual inspection of the structures etc. The functional and structural impact was validated using a combination of nuclear magnetic resonance (NMR) spectroscopy, Hydrogen-Deuterium eXchange Mass Spectrometry (HDX-MS), photo-induced electron-transfer fluorescence correlation spectroscopy (PET-FCS) and functional assays”.

We agree with the reviewer that the abstract indeed does not reflect the chronology of events that led us to identify the communication hinge residues. This has been adjusted.

2. In the introduction, a figure would be helpful, even in the supplementary, to get familiar with the ABC transporter nomenclature and what it corresponds to structurally (ID1, X-loop, RecA domain). Otherwise, reading of the following paragraph is arduous for a non ABC-transporter expert: “In this subfamily, interdomain crosstalk between NBD and TMD is mainly attributed to the proximity of the conserved Q-9,10 and X-loop motifs^{11–13} of the NBDs with the coupling helices^{14–17} of the TMDs. In half transporters like BmrA with six transmembrane helices, these intracellular α -helical linkers connect transmembrane helices 2 and 3 as well as 4 and 5. Together with the cytosolic regions of the respective transmembrane helices these linkers constitute the intracellular domains 1 and 2, i.e. ICD1 and ICD2, respectively¹⁸. Upon assembly into functional transporter dimers, ICD1 interacts with the NBD within the same subunit (in cis), while ICD2 reaches over to the NBD of the opposing subunit (in trans), inserting into a groove on the NBD surface between the RecA and the α -helical subdomains^{12,19}”.

We thank the referee for this important comment and apologize for having assumed too much familiarity with ABC transporter nomenclature. To improve clarity, we have rephrased the relevant paragraph and added a new figure (Supplementary Fig. S1) to provide the necessary background.

3. I’m sorry to say that I don’t see how PET FCS adds anything to the argument. NMR is less perturbative and so is HDX-MS. Could the authors clarify what is the additional insight brought by this method?

We thank the reviewer for their candid statement and acknowledge that we should have made the additional value of this method for our study, and as a powerful addition to the biophysicists’ toolkit, clearer. PET-FCS detects and directly measures the reconfiguration time constants of specific protein structural elements probed by tailored, site-specific fluorophore/tryptophan reporter design, thereby avoiding complications of dual reporter labelling as imposed by e.g. FRET. Such reconfiguration time constants of specific protein conformational coordinates, i.e. in our case the subdomain motions of the ABC transporter NBD between residue S516 and N459, are not detectable by NMR and HDX-MS. Thus, our PET-FCS studies not only complement the other methods we used but provide additional insights into global, interdomain motions between RecA and α -helical subdomain. These motions are affected by both nucleotides and mutations in the hinge region, and crucially our PET-FCS results showed that mutation of W413 has dramatic,

allosteric effects on domain motions along the remote conformational coordinate S516-N459. PET-FCS thus provides a structural rationale for the observed loss of allosteric coupling we observed in functional assays at time and length scales inaccessible to the other methodologies.

4. Why is there no direct comparison between WT and hinge mutants using HDX-MS? In general, the global view presented of the HDX data does not clearly show whether it supports the statement that “hinge mutations affect nucleotide and substrate binding” and that “nucleotide and substrate binding affect the hinge region.” Could a figure highlighting representative peptides from the nucleotide-binding region and the substrate-binding regions be shown for WT vs. the W139F mutant? Similarly, could peptides from the hinge region be shown for the apo, doxorubicin-bound, and ADP vanadate-bound states?

We agree that a direct comparison between WT and W413F mutant proteins would be of great value. However, we believe a global comparison in this context would not be entirely appropriate for the following reasons:

The W413F mutation causes destabilization of the transporter, as shown by a lower melting temperature (~36 °C vs. ~43 °C for WT, see Supplementary Table S3), which can affect protein folding and dynamics. This leads to increased deuteration across various regions of the mutant, complicating direct comparisons. In fact, 39 out of 95 shared peptides show higher deuteration in the mutant (see new Supplementary Figures S9, S10, S11 and new Supplementary Table 5). These peptides are mainly found in the NBD or in the ICD, i.e. near W413 in the NBD and ICD (suggesting a local effect), though some, such as peptide 57–69, are more distal, possibly reflecting allosteric changes.

*HDX is highly sensitive to subtle structural changes, and global comparisons risk obscuring meaningful site-specific effects. We therefore analysed each protein separately in the apo and ligand-bound states (new Supplementary Table 5), focusing on ligand-induced changes without the confounding influence of the differential effects of mutant and WT on transporter dynamics. Nonetheless, we identified 54 peptides with comparable deuteration in WT and mutant apo states (only two showed higher deuteration in WT (see new Supplementary Table 5). In response to the reviewer’s suggestion, we generated a comparative figure (new Fig. 4C) using six representative peptides from the NBD and ICD with similar apo deuteration levels. These data show reduced protection in the MgADP*Vi state for the mutant versus WT, supporting our hypothesis about W413’s role in signal transmission through the transporter. Please note that peptides in the drug binding site in the transmembrane helices are either not resolved or show deuteration levels too low for confident interpretation.*

5. Maybe the authors should agree on a consistent terminology to describe the cluster of residues and its effects and stick to it throughout the paper. Terms like nexus, region, network, cluster—as well as long-range, allosteric, bi-directional, coupling, dynamic, etc.—are used interchangeably, leading to sentences like “with a long-range dynamic coupling network” that feel quite abstract.

We have now homogenized our terminology for the communication hinge throughout the paper.

6. There is a mistake in the color-coding of the Δ HDX data in Supplementary Figure 9: “more accessible and more protected” colors (red and blue) have been reversed compared to figure 3.

Thank you for noticing! This was due to an inverted legend, the HDX figures have been corrected.

Reviewer #2 (Remarks to the Author):

This manuscript describes a critical role for 3 residues that act as a sort of hinge between the NBD and TMD of a Type IV ABC transporter, facilitating communication between the domains. This interdomain communication is critical for coupling of ATP hydrolysis and substrate transport. The data presented is primarily for BmrA, but sequence conservation and some chemical shift perturbation data on the MsbA and LmrA NBDs suggest that this hinge may be functionally important in other Type IV ABC transporters. This is a dense paper with a wealth of data. The authors make a convincing case for the importance of this 3-residue hinge, and the potential role of this novel hinge in regulating coupling within the Type IV ABC transporters makes this of interest to a broad community. The manuscript also demonstrates the value of evaluating mutations and the role of individual residues in both full length protein and isolated domains, as well as the importance of integrating an array of experimental methods to characterize such complex regulation of protein function. However, the range of methods, mutants and types of information provided by the different experiments makes the paper very complex. Careful presentation of the data and explanation in the text to focus on the key results and distinguish primary data and conclusions from supporting data sets is necessary to make the manuscript comprehensible to the broad readership of Nature Communications.

We thank the reviewer for their positive response to our work and especially for acknowledging the importance of a multifaceted experimental approach.

My primary concern is that the main message gets lost at times in the large amount of data. This is particularly true when the figures are so small that they are impossible to read or see in a printed copy. The color coding is very helpful, but sometimes light-colored fonts over colored backgrounds are difficult to read. Enlarging the individual panels to make sure the key features of the data is critical. It would also be helpful to divide some of the figures to clearly delineate data on the isolated NBD versus the full-length transporter, or to distinguish different types of data reporting on distinct regions and timescales so that each point is made clearly.

To improve clarity, we have revised the text related to Figure 2 and split the original figure, now presented as Figures 2 and 3 in the main manuscript, with enlarged panels for the NMR data to enhance readability. Figure 2 now focuses on the bidirectional coupling between the Walker A lysine (K380) in the nucleotide-binding site and the communication hinge, as revealed by NMR spectroscopy. Figure 3 presents PET-FCS data probing the subdomain dynamics between the RecA and α -helical domains of the NBD. The table with the fitting parameters for PET-FCS (previously Supplementary table 4) has been integrated into the new Fig. 3.

The only figure containing data for both the isolated NBD and the full-length transporter is Figure 1. There, we used NMR on the isolated NBDs to identify key residues, which were then functionally validated in ATPase and transport assays using the full-length system. Since these data directly build on each other, we would politely request to keep this in one figure. To guide the reader, we have clarified this point more explicitly in the updated figure legend.

While the authors discuss the significant chemical shift perturbations (Fig. 1) or changes in peak position and lineshape (Fig. 2), there are some sites that have minimal chemical shift perturbation (Fig.1 and 2D RICD2, Fig. C iv). This is not a major concern in the context of the vast amount of data, but should be properly acknowledged. Particularly since RICD2 has minimal CSP in several experiments.

We agree and have made this clearer in the revised version of the manuscript.

Reviewer #3 (Remarks to the Author):

The authors investigated a possible allosteric transmission pathway in ABC transporters. They first identified three conserved residues (R389, W413, R414 in BmrA) that appear to connect the nucleotide binding site in the Walker A motif to intracellular domain 2 (ICD2), providing a potential bridge to the substrate binding site. In a second step, the authors used NMR (HSQC) spectroscopy to understand whether the resonances of the three residues respond to the addition of nucleotides, which they did, confirming a direct link between the Walker A motif and this hinge region. Using the full-length transporter, the authors found that drastic mutations of these residues severely impaired ATPase and transport activity. Interestingly, a W413Y/F mutation still showed significant ATPase activity but a severe loss of transport activity, suggesting that this mutation decouples the two tasks. They also showed that a mutation in one of the residues affects the NMR resonances of the others. In particular, mutations in the hinge also affect the resonances of K380, which is located in the Walker A motif. Using photo-induced electron transfer (PET) in combination with fluorescence correlation spectroscopy (FCS), the authors found that replacing W413 with a less bulky residue (F) drastically reduces the quenching amplitude, indicating a more rigid structure. Similar results were obtained using hydrogen-deuterium exchange (HDX) coupled to mass spectrometry, which also showed rigidification in the presence of a nucleotide transition state analogue. Finally, the authors investigated the effect of substrate-binding on the ¹⁹F-NMR resonances of fluorinated W413 in nanodiscs and concluded that substrate-binding also affects the line-width of this resonance, indicating an effect on the dynamics.

Overall, this is an interesting study that uses an impressive set of tools to elucidate the allosteric coupling between substrate and nucleotide binding in ABC transporters. Overall, I am positive, but I have a number of questions that the authors should address in a revised version of the manuscript. In particular, the authors don't make it easy to follow their argument throughout, partly because it remains very descriptive (as opposed to mechanistic) and partly because some terms are not properly introduced.

We thank the reviewer for their very positive response to our manuscript and especially for their patience regarding our initial lack of clarity in introducing certain concepts. We hope the revised version is now more accessible. To improve readability, we have streamlined the hinge nomenclature, provided more detail on the underlying hypothesis, and added an explanatory figure (Fig. S1) to clarify terminology and illustrate the conformational changes involved in the catalytic cycle of type IV ABC transporters. Additionally, we have expanded our interpretation of the results to address, among other points, the transfer of information from the Walker A motif to the hinge (see also our response below regarding the Walker A K380A mutation).

The fact that a mutation of K380 affects residues in the hinge could be explained by a change in the dipole of the helix. Could this be a way of establishing communication between the nucleotide binding site and the hinge? The authors remain rather vague about the actual mechanism of communication.

The reviewer touches on a very interesting point. ABC transporter NBDs contain a short N-terminal loop followed by a helix ('Walker A helix') that harbor the residues belonging to the Walker A motif. In BmrA these residues are ³⁷⁶SGGG³⁷⁹ in the disordered loop and K380 as the most N-terminal residue of the α -helix.

Architecturally, we need to make the important point that in contrast to e.g. Serine proteases, where the helix dipole is oriented directly to the functional group thereby influencing its reactivity, the Walker A lysine sidechain is flexible in the absence of nucleotides. In line with this, our NMR data on the BmrA WT NBD show that in the apo state, the Walker A lysine backbone amide, as well as all neighboring residues in the loop and the helix are dynamic, resulting in severe signal broadening (see Supplementary Fig. 2B, previously Supplementary Fig. 1B). Nucleotide binding stabilizes the Walker A residues, including K380, presumably resulting in the conformational freezing of the Walker A helix, which will in return also affect residue R398 at the C-terminal end of the Walker A helix and a part of the communication hinge. Nucleotide binding also changes the local electrostatics by introducing negatively charged phosphate groups, thereby additionally changing the electrostatic properties of the Walker A helix.

In the case of the Walker A K380A mutant, the removal of a positively charged sidechain can likewise influence local electrostatics, helix dipole and H-bond strength in the helix. In addition, the α -helical propensity of alanine is slightly more pronounced than that of lysine, which may result in an additional small contribution to helix stabilization, although we cannot see the resonances belonging to the Walker A residues in spectra of the K/A mutant.

We have added a paragraph in the appropriate places in the results and discussion sections to extend on the mechanistic model of our findings.

2.) Since the authors don't say, do the NBDs dimerise in exactly the same structure as the full-length dimer (in BmrA)?

Advantageously, in most type IV ABC transporters including BmrA, the isolated NBDs remain monomeric in solution but retain nucleotide-binding capability, indicating structural and functional integrity. This simplifies the study of nucleotide interactions and makes the isolated NBDs valuable tools to generate hypotheses that can then be tested in a targeted manner in the full-length system as shown in our study. We have now explicitly commented on the monomeric state of the NBD in the introduction.

3.) I had problems with what to make of the timescales in the PET-FCS experiment. It seems to me that these timescales are extremely fit sensitive (looking at the noise in the correlation functions). However, what is very clear from these experiments is that the amplitude drops significantly when W413 is replaced by phenylalanine.

The standard errors of time constants from data fits are provided in the main text of the manuscript. They are reasonably low, i.e., between 10%-20%. However, we agree, that the fast ns- μ s time scale suffers from low signal-to-noise. This is a problem inherent to FCS because of the low flux of fluorescence photons on this very fast time scale in single-molecule experiments. Nonetheless, as the reviewer rightly notes, the change of amplitude of the relaxation in response to mutation W413F is dramatic and exceeds the error margins, as many of the determined time constants do.

4.) Related to the above point, I don't understand what is meant by three modes of action. Do the authors mean three structurally distinct states that interconvert on nano- to microsecond timescales? I think it is the lack of a picture of a potential mechanism that leads to the rather descriptive style of the paper. Perhaps the authors could be clearer about what they actually mean by modes of motion.

The interpretation of PET-FCS data is based on a two-state model of conformational change of the probed conformational coordinate (i.e., emission of fluorescence by the attached fluorophore or its quenching by a tryptophan residue that comes into proximity). Due to the placement of the fluorophore/tryptophan reporter pair at residues S516 and N459 in the two subdomains of the NBD, which were mutated to cysteine to attach the fluorophore and Trp to act as a quencher, respectively, this describes the motions between RecA and α -helical domain. Such interdomain fluctuations can include breathing, bending or shearing motions.

Kinetics of two-state conformational changes follow single-exponential behaviour. Data that can be described by a sum of n independent exponentials may be interpreted to report on n motions along n conformational coordinates of the probed segment. However, we cannot assign a single exponential in an n -exponential decay to a specific motion, i.e. the individual contributions of above-mentioned breathing, bending or shearing motions. Nonetheless, and this is the main finding here, which is inaccessible to other methods such as NMR and HDX-MS, we observe very clear differences in these combined interdomain motions in the presence of nucleotides or when the hinge is mutated. This allows us to unambiguously conclude that both nucleotides and the hinge affect the conformational changes between the two NBD subdomains. Since the groove formed between the two subdomains contacts the transmembrane domain, and because

mutations in the hinge interfere with NBD/TMD coupling, we infer that the underlying structural explanation stems from the way the RecA and α -helical domain move relative to each other.

We have now added a section to the main text to make this point clearer.

5.) More information is needed for the HDX experiments. Which exchange regime was used, EX1 or EX2? Can the deuterated fractions be compared between wt and phi mutant or can we only compare the differences between apo and holo?

The general conditions for HDX experiments are now reported in the new Supplementary Table 4, and a comparison of WT and mutant for individual peptides is presented in the new Supplementary Table 5. The uptake plots for HDX in the three conditions, Apo, ATP/Vi and doxorubicin for all the peptides of the WT and the W413F mutant have been included in new supplementary figures 10, 11 and 12, respectively.

We observed classic EX2 regime for the majority of the peptides, but notably some do display an EX1 regime, e.g. in ICD1 (between residues 108 to 132), ICD2 (between residues 189 to 255), the linker region between the TMD and the NBD (between residues 304-343) and the alpha-helical subdomain of the NBD (between residues 480-500). The example for EX1 in ICD2 is illustrated as a new supplementary figure 12 for the peptide 216-236 from WT and W413F mutant in the three tested conditions.

Regarding the direct comparison between the WT and the W413F mutant, we have now added a new Supplementary Table 5, as well as a new Fig. 4C. For details, please also see our answer to reviewer 1 regarding this point.

6.) On page 7, lines 181 and 186, the RWA residue is referred to as R398, but it appears that R389 is correct.

Thank you for pointing that out. This has been corrected.

7.) On page 6, line 194, the authors refer to an outward-facing state that hasn't been introduced before, causing some confusion. Perhaps this nomenclature is clear to anyone working on ABC transporters. It wasn't to me. I had to search the manuscript until I found Supplementary Figure 10, which explains that the transporter was crystallised in two forms. The authors would like to introduce this conformational flexibility much earlier, ideally in the introduction, as the interpretation of some of the results depends on it.

We realize we overlooked the need to introduce the structural transitions of ABC transporters clearly and early in the manuscript, and we appreciate the reviewer pointing this out. The transition between the inward-facing (IF) and outward-facing (OF) conformations has now been introduced in the introduction and we have added a new Supplementary Fig 1.

8.) Related to this point, the authors never specify which conformation or mixture of conformations the full-length transporter is in and how this equilibrium is controlled (is it controlled?). For example, in Figure 3A I had to conclude that the equilibrium is controlled by the nucleotide analogue.

*In the apo state (without nucleotides), the transporter will be in an equilibrium between an inward and outward facing state (see new Supplementary Fig. 1) but strongly favouring the inward facing state. In the presence of ATP, the NBDs dimerize, resulting in the preferential adoption of an outward facing state. However, due to ATP hydrolysis, this state is transient. Therefore, it can be stabilized by nucleotide and transition state analogues, such as MgADP*Vi. Here, the*

transporter is presumed to cycle through one round of ATP hydrolysis, expelling the released gamma phosphate and replacing it with a vanadate in a pentavalent state, thereby mimicking the transition state of hydrolysis. In this state, the transporter will be in the outward open state. This has now been explained in the text.

9.) On p. 7, line 223, the authors state that Fig. 2B shows HSQC spectra. However, Figure 2B shows ¹⁹F NMR spectra.

Thank you for pointing that out. This has been corrected.